


**Flower litters of alpine plants affect soil nitrogen and phosphorus rapidly in the eastern Tibetan Plateau**
AUTHORS:
Jinniu Wang[1, 2]
1. Chengdu Institute of Biology, Chinese Academy of Sciences/Key Laboratory of Mountain
Ecological Restoration and Bioresource Utilization, Chinese Academy of Sciences/Ecological
Restoration Biodiversity Conservation Key Laboratory of Sichuan Province, Chengdu 610041, China
2. International Centre for Integrated Mountain Development (ICIMOD), G.P.O. Box 3226, Kathmandu,
Nepal
Bo Xu[1, 3]
1. Chengdu Institute of Biology, Chinese Academy of Sciences/Key Laboratory of Mountain
Ecological Restoration and Bioresource Utilization, Chinese Academy of Sciences/Ecological
Restoration Biodiversity Conservation Key Laboratory of Sichuan Province, Chengdu 610041, China
3. University of Chinese Academy Sciences, Beijing 100049, China
Yan Wu[1]
1. Chengdu Institute of Biology, Chinese Academy of Sciences/Key Laboratory of Mountain
Ecological Restoration and Bioresource Utilization, Chinese Academy of Sciences/Ecological
Restoration Biodiversity Conservation Key Laboratory of Sichuan Province, Chengdu 610041, China
Jing Gao[1, 3]
1. Chengdu Institute of Biology, Chinese Academy of Sciences/Key Laboratory of Mountain
Ecological Restoration and Bioresource Utilization, Chinese Academy of Sciences/Ecological
Restoration Biodiversity Conservation Key Laboratory of Sichuan Province, Chengdu 610041, China
3. University of Chinese Academy Sciences, Beijing 100049, China
Fusun Shi (corresponding author)
Chengdu Institute of Biology, Chinese Academy of Sciences/Key Laboratory of Mountain Ecological
Restoration and Bioresource Utilization, Chinese Academy of Sciences/Ecological Restoration
Biodiversity Conservation Key Laboratory of Sichuan Province, Chengdu 610041, China
Email: shifs@cib.ac.cn
Tel: +86-28-82890537
Fax: +86-28-82890288



**Abstract**


Litters of reproductive organs have been rarely studied, despite their role in allocating
nutrients for offspring reproduction. This study determines the mechanism through
which flower litters efficiently increase the available soil nutrient pool. Field
experiments were conducted to collect plant litters and calculate biomass production
in an alpine meadow of the eastern Tibetan Plateau. Carbon, nitrogen, phosphorus,
lignin, cellulose, and their relevant ratios of litters were analyzed to identify their
decomposition features. A pot experiment was performed to determine the effects of
litter addition on soil nutrition pool by comparison between the treated and control
samples. Litter-bag method was used to verify decomposition rates. The flower litters
of phanerophyte plants were comparable with non-flower litters. Biomass partitioning
of other herbaceous species accounted for 10%–40% of the aboveground biomass.
Flower litter possessed significantly higher N and P levels but less C/N, N/P, lignin/N,
and lignin and cellulose concentrations than leaf litter. Flower litter fed soil nutrition
pool more efficiently because of their faster decomposition rate and higher nutrient
contents. Litter-bag experiment confirmed that the flower litters of *Rhododendron*
*przewalskii* and *Meconopsis integrifolia* decomposes approximately three times faster
than mixed litters within 50 days. Moreover, the findings of the pot experiment
indicated that flower litter addition significantly increased the available nutrient pool.
Flower litter influenced nutrition cycling in alpine ecosystems, as evident by its
non-ignorable production and significantly faster decomposition. The underlying
mechanism can enrich nutrients, which return to the soil, and non-structural
carbohydrates, which feed and enhance the transitions of soil microorganisms.
**Key words** alpine ecosystem, flower litter, chemical property, decomposition rate,
nitrogen, phosphorus





The growth and health of plants in their life history have been considerably influenced
by variations in the physical, chemical, and biological properties of soil, particularly
around the rhizosphere, although soil properties can also be mediated by plants. Plant
properties directly affect the productivity and function of an ecosystem (Chapin et al.,
1986; Chapin, 2003; Berendse and Aerts, 1987; Grime, 1998). In a natural
environment, plants continuously lose N and P in their whole life history and even
during litter production and decomposition (Laungani and Knops, 2009; Richardson
et al., 2009). N is a major constituent of several important plant substances (Vitousek
and Howarth, 1991). Most plants absorb N through soil compounds to support their
growth. In addition to the mineralization of soil organic matter, the decomposition of
plant residues can supply available N to plants and microorganisms. Similar to
nitrogen, P is closely associated with numerous vital plant processes. Nevertheless, in
most circumstances, P is limited because of its small concentration in soil; this
element is released slowly from insoluble P but is highly demanded by plants and
microorganisms (Bieleski, 1973; Richardson et al., 2009). As decomposition is a
prolonged process, plants contain concentrated nutrients comparable with soil, which
have significant effects on the biogeochemical cycle and feedbacks of plant–soil
interaction. However, these nutrients cannot be simply absorbed again to the soil
nutrient pool supplied by plants and microorganisms (Bieleski, 1973; Berendse and
Aerts, 1987).
In cold life zone ecosystems, plant biomass production is limited by N (Körner,
2003). Litter tends to be recalcitrant in cold environments (Aerts, 1997). In addition,
N is a key factor that determines the outcome of interspecific competition in
temperate-zone ecosystems (Laungani and Knops, 2009). Several studies reported that
litter can mediate the interactions between neighboring plants in infertile communities
(Nilsson et al., 1999, Xiong and Nilsson, 1999). In a succulent desert ecosystem in
Africa, fertile islands are formed in nutrient enrichment zones beneath shrubs; this
formation is attributed to a range of interactions between physical and biotic
concentrating mechanisms (Stock et al., 1999). In China, an experiment performed in
an alpine meadow ecosystem, the eastern Tibetan Plateau, indicated that soil N



availability and supply rates, as well as microbial biomass, can be enhanced by
*Stellera chamaejasme* L., which is an unpalatable poisonous weed that seriously
deteriorated the local rangeland (Sun et al., 2009). Another study in the gully region
of the Loess Plateau demonstrated that black locust improves most soil properties
(Qiu et al., 2010). Plants enhance the microbial immobilization of N when they
provide C to soil microorganisms. The nature of litter determines its palatability to
soil organisms, thereby influencing their composition and activity levels. Furthermore,
a few apparent effects of N may be caused by the low levels of polyphenols, which is
associated with high N concentrations in litter (Haynes 1986). The rate of decay and
concentrations of nutrients in the litter determine the rate of nutrient release, which
creates a positive feedback to site fertility. Hence, the chemical properties of litters
from different plant organs and their correlations with decomposition rate must be
determined.
Although inflorescences comprise only a small fraction of plant biomass and
production in Arctic and alpine vegetation, the inflorescence production can be a
significant proportion of the total production of species under certain special
circumstances (Mart ńez–Yr źar et al., 1999, Fabbro and K örner, 2004; Wookey et al.,
2009). Reproductive tissues present chemical composition that differs from vegetative
parts, resulting in a markedly faster decomposition and nutrient release, with
repercussions on nutrient cycling and patchiness (Buxton and Marten, 1989; Lee et al.,
2011). High contents of N and P exist in the reproductive organs of plants probably
because of their essential roles in plant growth and formation (e.g., high protein
content). Alpine ecosystems are thermally restricted and characterized by a low
material turnover rate (K örner, 2003). In a high altitude region, plants grow in a harsh
habitat that restricted their effective utilization of resources; in this regard, the total
available resource is less compared with that of plants in other regions (Fabbro and
K örner, 2004; Hautier et al., 2009). In long-term evolution, the allocation of
accumulated carbohydrates to reproduction is an adaptation strategy, leading to the
partitioning of reproductive organs, that is, the availability and timely mobilization of
adequate resources from the vegetative plant body to reproductive structures (Arroyo



et al., 2013). Thus far, probably due to reproductive organs' comparatively minor
biomass production and difficult to be collected, studies on their decomposition have
been limited particularly compared with those on leaf and other vegetative organs.
A fast decay of N-rich litters suggests that litter decay rates increase with increasing N
content. The initial rate of nutrient release is positively correlated with the initial
concentrations of N or P (MacLean and Wein, 1978; Aber and Melillo, 1980; Berg
and Ekbohm, 1983; Yavitt and Fahey, 1986; Stohlgren, 1988). In agricultural systems,
addition of fresh residues can stimulate the decomposition and net release of N from
indigenous soil organic matter (Haynes, 1986; Scott et al., 1996). Long-term increases
in N availability have also been reported following the additions of C to forests
(Groffman, 1999). Recently, a common-garden decomposition experiment in a wide
range of subarctic plant types demonstrated that structural and chemical traits are
better predictors for several high-turnover organs than structural traits alone (Freschet
et al., 2012). Decomposition rate of plant litters slightly differ because of their
species-specific traits and various organs, whose chemical qualities vary in a wide
range of plant types and environments. Thus, field investigation, pot experiment of
litter addition, and litter-bag experiment were conducted in this study to address the
following:
1) Should decomposition of flower litter be considered according to inflorescence
biomass production, and/or allocation?
2) What are the unique chemical properties of flower litters that influence their faster
decomposition rate compared with leaf litters?
3) Is pulsed effect evident on soil available N and P particularly in special temporal
period and spatial location as determined through pot experiment?
**Materials and Methods**
*Study area*

145        The field site is located at the foot of Mt. KaKa, which belongs to the middle

section of Minshan Mountain, eastern Tibetan Plateau (**Fig. 1**), with a mean annual
precipitation of 720 mm. More than 70% of precipitation falls in summer from June to
August. Snowfall usually occurs from the end of September to the next early May.



Vegetation presents a typical alpine meadow with numerous and unique alpine plants.
Mosses are abundant and cover most of the ground. The moss layer is dominated by
*Polytrichum swartzii* and *Trematodon acutus* c. mull. Vascular plants include species
mainly belonging to *Kobresia* and *Carex*. Other common species are *Festuca,*
*Gentiana*, and *Leontopodium.* Plant roots in this ecosystem are generally confined to
the surface A-horizon (2–20 cm). A few dwarf shrubs are scattered sporadically in the
meadow, e.g., *Rhododendron* and *Salix*. The soil type is dominated by Mat Cry-gelic
Cambisols (i.e., silty loam inceptisol, *Chinese Soil Taxonomy Research Group*, 1995).
*Sampling*
During the blooming period from the end of May until mid-June and from the end
of July until early August, flower litters of 14 earlier flowering plants species and 15
later flowering plants species were carefully collected in 2012 at two sites, namely,
Mt. KAKA (103°42′ E; 32°59′ N, 3500–3900 m a.s.l.) and Bow Ridge Mountain
(103°42′ E; 33°1′ N, 3600–3850 m a.s.l.). These species were tentatively classified
into five groups according to Raunkiaer's life-form system (i.e., chamephyte,
geophyte, hemicryptophyte, phanerophyte, and therophyte; **Table 1**). Mixed litter of
alpine meadows were sampled on the Mt. Kaka (3950 m. a.s.l.), and leaf litters of 13
dominant species were also collected to compare their chemical properties with
flower litters. Both types of litters were first spread on blotting paper for air drying. A
small portion of each litter was further dried in an oven for 48 h to calculate dry
matter content.
*Experimental design*
Polyvinyl chloride (PVC) pots (15 cm deep, 20 cm diameter at the top, and 12 cm
diameter at the bottom) were filled with 2 kg of soils, which were collected in autumn
of 2011. The collected soil samples were stored at 4 ℃. The samples were sieved
through 2 mm mesh and then mixed thoroughly. The soil surface of each treatment
was added with 5 g of flower litters or mixed litters (calculated as dry weight) on June
21 (14 species, earlier flowering plants) and Aug 11, 2012 (15 species, later flowering
plants). The surface was covered with a thin layer of soil to avoid being blown by
wind. Other two additional treatments were conducted without litter addition (control)



and with mixed leaf litter addition, respectively. In total, the pot experiment consisted
of 33 treatments with three replicates, with a total number of 99 pots. All of the pots
were carefully buried 12 cm deep into the field to maintain the same soil temperature
in the experimental field. The pots were randomly distributed, and their top edges
were approximately 3 cm above the ground to prevent runoff from outside. All of the
pots were rearranged every week to create a similar microclimate. After 50 days, soil
samples were separately obtained from the PVC pots and mixed evenly by sieving
through a 2 mm mesh. Soil samples were collected from three points of each pot in
the center and then mixed to avoid the boundary layer effect. The samples were stored
in an ice box prior to chemical determination.
*Decomposition rate*
A litter bag with a size of 14 cm × 20 cm was used to determine the
decomposition rate of different plant litters. The bag was double faced and made from
nylon net material with above (4.5 mm × 4.5 mm mesh) and below layers (0.8 mm ×
0.8 mm mesh). The above layer with bigger mesh size allowed free access for most
micro-arthropods, which dominate the soil fauna of alpine meadow in the eastern
Tibetan Plateau, whereas the below layer with smaller mesh size can reduce litter
spillage from the litter bags in the process. As representative species, flower litters of
*R. przewalskii* and *M. integrifolia* and mixed litter were packed into litter bags with
the edges sealed on June 21, 2012. The litterbag experiment was conducted to
compare the decomposition rate of flower litters and mixed litter. Each treatment had
eight replicates. After 7 weeks (August 8, 2012), litter was obtained from the litter
bags and dried in an oven for decomposition calculation. Litter decomposition rates
can be determined by the following equation.
**DR = (P−R)/P × 100**
where *DR* is the decomposition rate, *P* is primary litter mass in the litter bags, and *R*
refers to residue litter before determining percentage mass loss.

*Soil chemistry determination*
Total dissolved N (TDN) contents were determined using unsieved fresh moist




soil subsamples. Soil subsamples were extracted using 2 M KCl and shaken for 1 h at
room temperature (20 ℃), with a soil-to-solution ratio of 1:5 (weight/volume). The
extracted solution was filtered through filter paper before further determination (Jones
et al., 2004). $NH_4^+$-N and $NO_3^-$-N were analyzed with the indophenol blue
colorimetric (Sah, 1994) and ultraviolet spectrophotometry methods (Norman et al.,
1985), respectively. Dissolved organic nitrogen (DON) was calculated by subtracting
dissolved inorganic N ($NH_4^+$-N, i.e., DHN and $NO_3$-N, i.e., DNN) from TDN. Soil
solutions were extracted by centrifugal drainage, whereas the exchangeable pool was
extracted with 2 M KCl by using the methods reported by Jones et al. (2004). Total
phosphorus (TP) and A-P in soils were estimated by extraction with 0.5 M sodium
hydroxide sodium carbonate solution (Dalal, 1973). Microbial biomass carbon (MBC)
and microbial biomass nitrogen (MBN) contents were determined through the
chloroform–fumigation direct-extraction technique. Correction factors of 0.54 for N
and 0.45 for C were used to convert the chloroform labile N and C to microbial N and
C (Brookes et al., 1985).
***Data analysis***
One-way ANOVA was applied to compare values between the treatments and the
control. Post-hoc multiple comparisons were adopted when the groups were three or
more. Multivariate ANOVA was conducted to determine the effects of blooming time
and different addition of litters and their interactions. To simplify the comparison of
soil N and P between control (without flower litter) and the treated (with flower litter),
we defined an index $\alpha$ as: $\alpha = Ln (N_2/N_1)$. $\alpha > 0$, $N_2 > N_1$; $\alpha < 0$, $N_2 < N_1$; $\alpha = 0$, $N_2 =$
$N_1$. $N_1$ is the control treatment without flower litter, and $N_2$ indicated nutrition value
(N or P) of flower litter treatment. Descriptive analysis was operated to demonstrate
$\alpha$ values of different N and P fragments in various species litters addition treatment.
The box plots provide the distribution of the values by the medians (central line), the
quartiles 25% and 75% (box), and the ranges (whiskers) of ratios. Differences were
tested at $P < 0.05$ by using Tukey multiple range test in SPSS 19.0 software package
(SPSS Inc., Chicago, IL, USA). The normality of data was tested with one-sample
K-S test and Q-Q plot. Otherwise, log-transformation was adopted to meet the



normality requirement. Homogeneity of variance test was also utilized during the
analysis. In the figures and tables, information is presented as means and standard
errors of means. All of the differences were tested at the $P = 0.05$ level.
**Results**
**Inflorescence information of alpine plants from different life forms**
**Table 1** General description of flower litters in the study.

| | Life form | Size of inflorescence (cm) | Dominant (Y/N) | Color | Dry matter content (%) |
|---|---|---|---|---|---|
| *Caragana jubata* | C | 1-1.5 | N | white | 29.81 |
| *Primula orbicularis* | H | 1.5 | Y | yellow | 23.29 |
| *Potentilla anserina* | G | 1-1.8 | Y | yellow | 51.9 |
| *Rhododendron capitatum* | P | 2-3 | Y | purple | 32.84 |
| *Viola rockiana* | H | 1 | N | yellow | 25.22 |
| *Myricaria squamosa* | P | 0.5-1 | N | pink | 30.95 |
| *Potentilla saundersiana* | G | 1-1.4 | N | yellow | 54.01 |
| *Taraxacum lugubre* | H | 3-4 | Y | yellow | 14.97 |
| *Aster tongolensis* | H | 4-5 | N | blue | 28.72 |
| *Cardamine tangutorum* | G | 0.8-1.5 | N | lavender | 13.08 |
| *Spiraea alpina* | P | 0.5-0.7 | Y | fallow | 32.58 |
| *Caltha scaposa* | H | 3-4 | Y | yellow | 30.43 |
| *Rhododendron przewalskii* | P | 4-5 | Y | pink | 33.33 |
| *Meconopsis integrifolia* | H/T | 5-7 | N | yellow | 21.79 |
| *Stellera chamaejasme* | C | 0.5 | N | red | 28.11 |
| *Potentilla fruticosa* | P | 2-3 | Y | yellow | 30.43 |
| *Meconopsis punicea* | H/A | 5-8 | N | red | 33.57 |
| *Meconopsis violacea* | H | 4-6 | N | purple | 35.70 |
| *Sibiraea angustata* | P | 0.8 | Y | white | 29.50 |
| *Polygonum macrophyllum* | H | 0.2 | Y | pink | 21.79 |
| *Pedicularis megalochila* | C | 0.8-1 | N | red | 33.57 |
| *Ligularia virgaurea* | C | 1.5 | N | yellow | 16.78 |
| *Pilose Asiabell* | C | 2-2.5 | N | pale green | 22.26 |
| *Oxytropis ochrocephala* | C | 1 | N | fallow | 28.72 |
| *Pedicularis longiflora* | C | 0.8 | N | yellow | 28.11 |
| *Hedysarum vicioides* | C | 1 | N | pink | 30.02 |
| *Gentiana sino-ornata* | C | 3-5 | Y | purple | 44.10 |
| *Leontopodium sinense* | C | 0.2-0.5 | Y | white | 56.92 |
| *Cremanthodium lineare* | G | 1.2-1.7 | Y | yellow | 48.93 |

Note: C, H, G, P, and T represent chamaephyte, hemicryptophyte, geophyte (one of subdivided groups in Cryptophytes),
phanerophyte, and thermophile, respectively. Y and N indicate whether the species is dominant or not in the community.
Twenty-nine species were reported in this study, and these species were divided into



two groups based on blooming time, that is, earlier flowering species and later
flowering species. According to Raunkiaer's life-form system, earlier flowering
species mainly consisted of hemicryptophyte, geophyte, and phanerophyte, whereas
more than half of later flowering species comprised chamaephyte. Nearly half of the
tested species were dominant or co-dominant in their respective communities. The dry
matter content of flower litters in all of the species was ranked from 10% to 60%.
Various sizes of inflorescence were distributed from 0.2 cm to 8 cm in diameter.
**Flower litter production of dominant species and their biomass allocation**

256        Among 13 dominant species, the flower litters of phenerophyte plants, whose

flower litters are comparable with non-flower litters, were calculated through
comparison with non-flower litters during the flower litter collection (**Fig. 2 (a)**). The
dry weights of flower litters per unit area were 10–40 g m$^{-2}$, whereas their non-flower
litters were only 5–25 g m$^{-2}$. Although neither of the flower litters of *S. angustata* nor
*R. capitatum* were significantly different compared with their non-flower litters ($P >$
0.05), the difference between the two remained noticeable, whose values were 28.03 $\pm$
3.56 g m$^{-2}$ versus 13.21 $\pm$ 1.49 g m$^{-2}$ for *R. capitatum* and 19.58 $\pm$ 3.50 g m$^{-2}$ versus
12.95 $\pm$ 0.61 g m$^{-2}$ for *S. angustata*, respectively. The production of flower litters was
higher than that of non-flower litters. The other three species significantly produced
more flower litters than non-flower litters (*R. przewalskii*: $F = 15.76$, $P < 0.001$; *P.*
*fruticosa*: $F = 4.76$, $P < 0.05$; *S. alpine*: $F = 10.18$, $P < 0.01$). The flower litters of the
eight herbaceous species were compared with their individual aboveground biomass
(**Fig. 2 (b)**), which ranked from 10% to nearly 40%. This finding indicated that flower
litter should be considered to determine the effect of plants on soil nutrition pool
during growing season.




**Comparison of chemical properties between flower and leaf litters**

Total C content was not significantly different between flower and leaf litters (**Fig. 3 (a)**, $F = 1.80$, $P = 0.199$). However, the levels of cellulose, lignin, and structure C of leaf litter were significantly higher than those of flower litter ($F = 6.74$, $P < 0.05$; $F = 5.77$, $P < 0.05$; $F = 10.99$, $P < 0.01$). Hence, flower litter probably contains more non-structure C than leaf litter.

Both N and P contents of flower litters were significantly higher than those of leaf litters (**Fig. 3 (b)**). N in flower litters was nearly doubled to that of leaf litter ($23.17 \pm 1.52$, $11.87 \pm 0.77$; $F = 45.70$, $P < 0.001$). More than twice the amount of P were also present in flower litters compared with that in leaf litters ($2.95 \pm 0.25$, $1.12 \pm 0.12$; $F = 43.87$, $P < 0.001$).

For the implication of the ratio of different chemical properties, C/N, N/P, and lignin/N were determined to compare flower and leaf litters. All the three indicators of leaf litter were significantly higher than those of flower litters (**Fig. 3 (c)**). As parameters used to demonstrate decomposition rate, C/N and lignin/N of leaf litter were nearly double to those of flower litter ($39.27 \pm 4.16$, $19.80 \pm 1.39$, $F = 37.78$, $P < 0.001$; $21.09 \pm 2.25$, $12.79 \pm 1.15$, $F = 7.91$, $P < 0.01$). Furthermore, N/P of flower litter was significantly higher than that of leaf litter ($8.42 \pm 0.42$, $11.60 \pm 0.56$; $F = 20.62$, $P < 0.001$). These findings indicated that flower litter can supply more P per unit N than leaf litter.

294

295

296





**Effects of flower litter on different fragments of soil nitrogen pool**

Earlier flowering species exerted positive effects on soil DIN, DON, TN, DNN, and

DHN (**Fig. 4 (a)**), with the addition of their flower litters according to their size of $\alpha$

values. Most parameters were higher than 0, which indicated that $N_2 > N_1$. Flower

litter increased soil N pool. All of the minimum $\alpha$ values of five indices were also

higher than 0 (**Table 2**, 0.42–1.29), which indicated that flower litter addition

significantly increased different fragments in soil N pool ($P < 0.001$). Among the later

flowering species, except *G. sino-ornata* and *L. sinense*, soil N indices were

significantly improved with flower litter addition, as demonstrated through $\alpha$ values

higher than 0 (**Fig. 4 (b)**, **Table 2**). Later flowering species differed from earlier

flowering species, with minimum $\alpha$ values lower than 0, which resulted from the

exceptions of *G. sino-ornata* and *L. sinense*. However, all of the mean $\alpha$ values were

higher than 0, which presented general results after flower litter addition (0.36–1.49).

Different fragments of soil N pool were significantly enhanced only after 50 days ($P <$

0.001). Interactions between flowering time and litter addition for DIN, DNN, and

DHN were significant ($F = 5.043$, $P < 0.05$; $F = 7.947$, $P < 0.01$; $F = 24.143$, $P < 0.05$,

respectively) but not for TN and DON ($F = 0.470$, $P = 0.496$; $F = 2.798$, $P = 0.100$,

respectively). Flower litters from the two categories of plants with different flowering

times were processed and compared in the addition experiment; as such, the effects of

both factors and their interactions were evaluated. Different flowering times

significantly affected DIN, DNN, and DHN (**Table 3**, $P < 0.01$) but did not

significantly influences DON and TN ($F = 0.47$, $P = 0.50$; $F = 2.80$, $P = 0.10$,

respectively). As illustrated in **Fig. 4**, litter addition had significant effects on all of

the N fragments, which was in accordance with the results in **Table 3**. The interaction

of flowering time and litter addition exerted similar effects on different N fragments

in soil with flowering time solely.



**Table 2** α values of different nitrogen fragments in various species litters addition
treatment (n = 14 and n = 15 in earlier flowering species and later flowering species,
respectively).

| Flowering period | Index | Mean | Std. Error | Minimum | Maximum | *F* | *P* |
|---|---|---|---|---|---|---|---|
| Earlier flowering | DIN | 1.66 | 0.07 | 1.07 | 2.20 | 578.88 | 0.000 |
| | DON | 1.76 | 0.26 | 0.67 | 4.46 | 46.45 | 0.000 |
| | TN | 1.67 | 0.06 | 1.29 | 2.05 | 719.05 | 0.000 |
| | DNN | 1.67 | 0.07 | 1.08 | 2.23 | 563.90 | 0.000 |
| | DHN | 0.97 | 0.12 | 0.42 | 2.06 | 68.25 | 0.000 |
| Later flowering | DIN | 1.07 | 0.17 | -0.63 | 1.50 | 40.29 | 0.000 |
| | DON | 1.49 | 0.29 | -0.18 | 3.29 | 27.04 | 0.000 |
| | TN | 1.29 | 0.21 | -0.37 | 2.40 | 38.37 | 0.000 |
| | DNN | 1.11 | 0.18 | -0.75 | 1.55 | 37.77 | 0.000 |
| | DHN | 0.36 | 0.05 | -0.09 | 0.72 | 60.64 | 0.000 |


**Table 3** Multifactorial analysis of variance for the effects of flowering time, litter
addition, and their interactions on different nitrogen fragments.

| Source of variation | DIN | | DON | | TN | | DNN | | DHN | |
|---|---|---|---|---|---|---|---|---|---|---|
| | *F* | *P* | *F* | *P* | *F* | *P* | *F* | *P* | *F* | *P* |
| *Corrected Model* | 31.74 | **0.00** | 23.62 | **0.00** | 59.25 | **0.00** | 69.24 | **0.00** | 54.07 | **0.00** |
| Flowering time | 5.05 | **0.03** | 0.47 | 0.50 | 2.80 | 0.10 | 7.93 | **0.01** | 24.36 | **0.00** |
| Litter addition treatments | 86.44 | **0.00** | 70.24 | **0.00** | 173.47 | **0.00** | 194.34 | **0.00** | 117.00 | **0.00** |
| Flowering time × Litter addition treatments | 5.05 | **0.03** | 0.47 | 0.50 | 2.80 | 0.10 | 7.93 | **0.01** | 24.36 | **0.00** |

Note: *P* values for significant effects and interactions are in bold.




**Effects of flower litter on soil TP and A-P**

**Table 4** α values of TP and A-P with flower litter added treatments (n=14 and n=15 in earlier flowering species and later flowering species, respectively).

| Flowering period | Index | Mean | Std. Error | Minimum | Maximum | *F* | *P* |
|---|---|---|---|---|---|---|---|
| Earlier flowering | TP | 0.02 | 0.03 | -0.04 | 0.08 | 8.498 | **0.007** |
| | A-P | 0.31 | 0.17 | 0.67 | 0.13 | 47.39 | **0.000** |
| Later flowering | TP | 0.03 | 0.11 | -0.20 | 0.12 | 0.97 | 0.33 |
| | A-P | 0.50 | 0.23 | 0.06 | 0.37 | 68.82 | **0.000** |

**Table 5** Multifactorial analysis of variance for the effects of flowering time, litter addition, and their interactions on TP and A-P.

| Source of variation | TP | | A-P | |
|---|---|---|---|---|
| | *F* | *P* | *F* | *P* |
| *Corrected Model* | 1.07 | 0.37 | 43.01 | **0.00** |
| Flowering time | 0.02 | 0.90 | 6.44 | **0.01** |
| Litter addition treatments | 3.17 | 0.08 | 114.14 | **0.00** |
| Flowering time ×Litter addition | 0.02 | 0.90 | 6.44 | **0.01** |

Note: *P* values for significant effects and interactions are in bold.

Flower litters exerted different effects on soil TP and A-P. Soil TP increased in treatment with early flowering litters (**Fig. 5**, **Table 4**, $F = 8.498$, $P = 0.007$) but not in later flowering litters. The minimum α values were lower than 0 (−0.04 and −0.20, respectively). However, A-P of both litter treatments was significantly positively stimulated ($F = 47.39$, $P < 0.001$; $F = 68.82$, $P < 0.001$), whose α values were both higher than 0 (0.67–0.13 and 0.06–0.37, respectively). Multifactorial analysis indicated that soil TP was not significantly different between treated with flower litter and control in general (**Table 5**, $F = 1.07$, $P = 0.37$). No significant interaction was evident between flowering time and litter addition treatments on soil TP ($F = 0.01$, $P = 0.93$). Litter addition treatments alone only had a marginal significant effect on soil TP ($F = 3.17$, $P = 0.08$). Moreover, both minimum α values were lower than 0, but TP was not significantly different between treatments with later flowering litters and control treatment ($F = 0.97$, $P = 0.33$), which mainly resulted from *G. sino-ornata*, *L. sinense*, and *C. lineare*. Nevertheless, A-P increased significantly after flower litter addition ($F = 43.01$, $P < 0.001$), with a significant interaction between flowering time and litter addition ($F = 6.44$, $P < 0.05$).



355

**Comparison of decomposition rate between flower litter and mixed litter**

Two typical plant species, which are widely distributed and easily collected, were assessed to compare the decomposition rate of flower litter and mixed litter. Differences in decomposition rate among flower litter of two species and mixed litter were supposed to be significant (**Fig. 6**, $F = 130.34$, $P < 0.001$). The flower litters of *R. przewalskii* and *M. integrifolia* decomposed greatly faster than mixed litter. However, within only 50 days, more than 20% of *R. przewalskii* and *M. integrifolia* flower litters decomposed, whereas the decomposition rate for mixed litter was approximately 6% (i.e., the former was nearly three times faster). Moreover, no significant differences were evident in the decomposition rates of the flower litter of *R. przewalskii* and *M. integrifolia* ($P = 0.371$).

**Discussion**

Plant litter decomposition is a critical step in the formation of soil organic matter, mineralization of organic nutrients, and C balance in terrestrial ecosystems (Austin and Ballaré, 2010). Species-specific variations in plant phenology can affect production of litter fall, which is noticeable during the growing season from the aspect of nutrient cycling although the peak of litter fall happens in autumn. Thus, the early litter fall of alpine plants during the study period from May to August can be a potential nutrient source when nutritional demands increase for rapid growth and development. In particular, the amount of flower fall in study area exceeds the leaf fall during the flowering season. A previous study indicated that reproductive litter production accounted for < 10% of the total litter in January–August and 13%–26% in September–December (Sanches et al., 2008), which was mainly triggered by rainfall variability that directly altered litter production dynamics and indirectly altered forest floor litter. In addition, the flowers are more nutritional than the leaves in terms of nutrients necessary for plant growth (Lee et al., 2011). In this study, summit production of flower litters are booming during special periods for both earlier flowering and later flowering species. Flower biomass of herbaceous plants accounts



for 10% to approximately 40% of total aboveground biomass. Moreover, these flower
litters produced considerably earlier than other aboveground litters that dropped at the
end of growing season. Furthermore, flower litters and non-flower litters (mainly
constituted of leaves) of woody plants were 10–40 g m$^{-2}$ and 5–25 g m$^{-2}$, respectively,
which clearly implies that flower litter can be a comparable decomposition substrate
in alpine ecosystems even for phenerophyte plants.
Litter production and decomposition are controlled by biological and physical
processes, such as the activity and composition of soil and litter fauna and climate
variations (Meentemeyer, 1978; Cornejo et al., 1994; Wieder and Wright, 1995; Aerts,
1997; Cleveland et al., 2004). An integration of index or traits has been recommended
to indicate process and rate of litter decomposition. Generally, tissues with high lignin,
polyphenol, and wax contents and higher lignin:N and C:N ratios exhibit slow
decomposition. The effect of litter quality on decomposition rates was extensively
discussed in the literature, and C/N and lignin/N ratios have been commonly accepted
as main explanatory factors (Melillo et al., 1982; Berg, 2000). Leaf litter with C/N
ratios lower than 30 is known to decompose easily and yield a mull humus type,
whereas C/N ratios above 30 result in N immobilization (Heal et al., 1997) and
decomposition retardation. In the present study, flower litter had significantly less
C/N ratio (19.80 ± 1.39, less than 30) than leaf litter (39.27 ± 4.16, more than 30).
Lignin content in flower litters was significantly less than that in leaf litters (211.37 ±
8.63 mg kg$^{-1}$ and 237.88 ± 6.89 mg kg$^{-1}$, respectively; $F = 5.77$, $P = 0.02$), similar to
cellulose (266.93 ± 4.92 mg kg$^{-1}$   and 283.75 ± 4.21 mg kg$^{-1}$, respectively; $F = 6.74$,
$P = 0.01$), which is one of the major cell-wall constituents. All of the results are in
accordance with previous studies. Decomposition rate is negatively correlated with
the concentration of lignin, which is a group of complex aromatic polymers that
serves as a structural barrier impeding microbial access to labile C compounds (Swift
et al., 1979; Taylor et al., 1989; Austin and Ballaré, 2010; Talbot and Treseder, 2012).
Moreover, greater non-structural carbohydrates existed in flower litters than those in
other litters, as indicated by the absence of significant differences of total C content
between flower litters and other litters. However, the structural carbohydrates of



flower litters were significantly less than that of leaf litters. This finding can be
inferred from the contents of lignin and cellulose (**Fig. 3 (a)**). Hence, flower litters
can promote nutrients that easily complement soil (Parton et al., 2007) for plants in
their whole life history. Decomposition rates of leaf litters have been considered
recently from their lignin/N or lignin/cellulose (Talbot and Treseder, 2012; Cornwell
et al., 2008). Furthermore, in the present study, lignin/N was less in flower litters
(almost 50% in leaf litters, i.e., $12.79 \pm 1.15$ and $21.09 \pm 2.25$, respectively), whereas
N/P was higher than that of leaf litters.
A litterbag experiment was adopted and confirmed that the decay rates of flower
litters were significantly faster than that of other litters, which is in accordance with
the fast decomposition of *R. pseudoacacia* flower from an experiment performed in
Korea (Lee et al., 2010). Flower litters contained significantly higher N and P
contents than leaf litters (**Fig. 3 (b)**). Plant litter available to the decomposer
community encompasses a broad range of issues that differ in chemical and physical
properties (Swift et al., 1979). P has to be highlighted because it has been regarded as
essential for a long time, which causes a limited attention on mechanisms that drive P
limitation and their interactions with the N cycle (Vitousek et al., 2010). Although soil
generally contains a large amount of total P, only a small proportion is immediately
available for plant uptake from the soil solution. In most soils, the concentration of
orthophosphate in solution is low (Richardson et al., 2009). P is derived mainly from
rock weathering and related biogeochemical cycle, and ecosystems begin their
existence with a fixed complement of P, and even very small losses cannot be readily
replenished (Walker and Syers, 1976). The present study indicated that decomposition
of flower litter can be one of the beneficial source of soil A-P in alpine ecosystems.
Nevertheless, the current study regarding the characteristics and driven mechanism of
this source remains at the first stage. Variation in soil physical-chemical properties,
vegetation types, and microbial activities can significantly affect chemical
compositions and forms and biological availability of soil P directly or indirectly.






**Table 6** Comparison medium values of soil solution pool and soil microbial biomass
between litter addition treated (flower litter and mixed litter) and control.

| Treatments | Soil solution pool (mg g$^{-1}$) | | Soil microbial biomass (mg kg$^{-1}$) | | |
| --- | --- | --- | --- | --- | --- |
| | Nitrate-N | Ammonium-N | MBC | MBN | MBC/MBN |
| Flower litter | 46.8 | 0.55 | 102.05 | 73.02 | 1.40 |
| Mixed litter | 32.4 | 0.45 | 68.08 | 69.29 | 0.98 |
| Control | 30.93 | 0.53 | 46.25 | 67.13 | 0.69 |

Decay rates of different plant organs reflect the diversity that fruits decompose faster
than leaves, which in turn decompose faster than woody plant parts (Swift et al., 1979;
Kögel–Knabner, 2002). Flower litters decompose rapidly with higher N and P levels
supplied to soil, particularly from nitrate-N in soil solution pool (**Table 6**). The soil
solution pool has been improved noticeably from 30.93 mg g$^{-1}$ to 46.8 mg g$^{-1}$ in
flower litter treatment compared with the control. Histogram for **α** values of DIN and
A-P also presented soil available nutrients positively stimulated by flower litter (**Fig.**
**7**) for their values distributed at an interval greater than 0. The high DOC values in
flower litter may influence N and P in soil through C substrate supplement for soil
microorganisms to enhance N immobilization. Recent empirical studies noted that the
changing microbial community composition significantly affects ecosystem processes,
such as litter decomposition (Strickland et al., 2009; Ramirez et al., 2012). Shifts from
bacterial-dominated to fungal-dominated decomposition happened over short (days to
a few months) periods (Poll et al., 2008; McMahon et al., 2005). Although the present
study did not present the precise analysis of microbial community, both MBC and
MBN differed greatly between different treatments (**Table 6**). Litter addition
increased them obviously, which is evident not only in microbial biomass C and N but
also in their C:N ratios (1.40, 0.98, and 0.69 for flower litter, mixed litter, and control,
respectively). Therefore, microbial community composition varied depending on
nutrient supplement from litters. Flower litter contains more than twice MBC
(increased from 46.25 to 102.05); hence, microbial biomass and their activities
enhance potentially.
Several unexpected species in the experiment reduced soil available nutrients



probably because their specific chemical properties, which change as a result of
microbial activities and nutrient dynamics (Karmarkar and Tabatabai, 1991), may
negatively affect soil microorganism biomass or activities (Wardle et al., 1998,
Cipollini et al., 2012). Furthermore, soil microbial communities can be modified
through time in response to allelopathic plants with known or potential effects on
plant communities (Cipollini et al., 2012, Inderjit and Weiner, 2001). Mineralization
and nitrification can be subdued by inhibitory compounds from the exudates of a
certain plant species, which come from a negative aspect and mainly result from
suppression of related microbes (Cipollini et al., 2012). In another positive
perspective, considering "*priming effect*" once flower litter is added in moderate
treatments causes strong short-term changes in the turnover of soil organic matter and
nutrient release follows litter decomposition (Jenkinson et al., 1985; Kuzyakov et al.,
2000; Blagodatskaya and Kuzyakov, 2008). Hence, N and P availability in the soil of
alpine ecosystem can be maintained in part by tissue chemistry favorable to microbial
decomposition and release of nutrients.
This study provides evidence that plant species, through tissue chemistry, biomass
allocation, and phenology, affect local soil properties in alpine ecosystem. Soil has
specific susceptibility to decomposition of biochemical compounds in plant tissues,
on a spectrum from quickly decomposed labile to relatively recalcitrant.
Decomposition rates can be markedly affected by particle size, surface area, and mass
characteristics (Angers and Recous, 1997). In addition, physical toughness (lignin, dry
matter content, or C content) can be suitable predictors of decomposition across all of
the organs. Structural (lignin, DMC) and chemical (N) traits together are proposed to
be better predictors for several high-turnover organs than structural traits alone
(Freschet et al., 2012). Flower litters have these intuitive benefits chemically and
physically, but physical components of litter quality have received little attention in
the research on litter quality. Future climate changes in temporal patterns are likely to
have important direct and indirect consequences on litter dynamics as well as on
phenology and decay process temporally and spatially. In brief, the question of the
essentiality and fundamentality of litter decomposition, especially under natural



conditions, remains unresolved although the key role of litter quality in decomposition
and in ecosystem function is generally clear.
**Acknowledgment**
This study was financially supported by the International Cooperation Project of
Science and Technology Department of Sichuan Province (2014HH0056), China
Postdoctoral Science Foundation (2014M552385), and   National Natural Science
Foundation of China (31400389). Authors would like to acknowledge the Key Lab of
Ecological Restoration and Biodiversity Conservation of Sichuan (ECORES) for their
support in laboratory facilities.




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





**Figures**

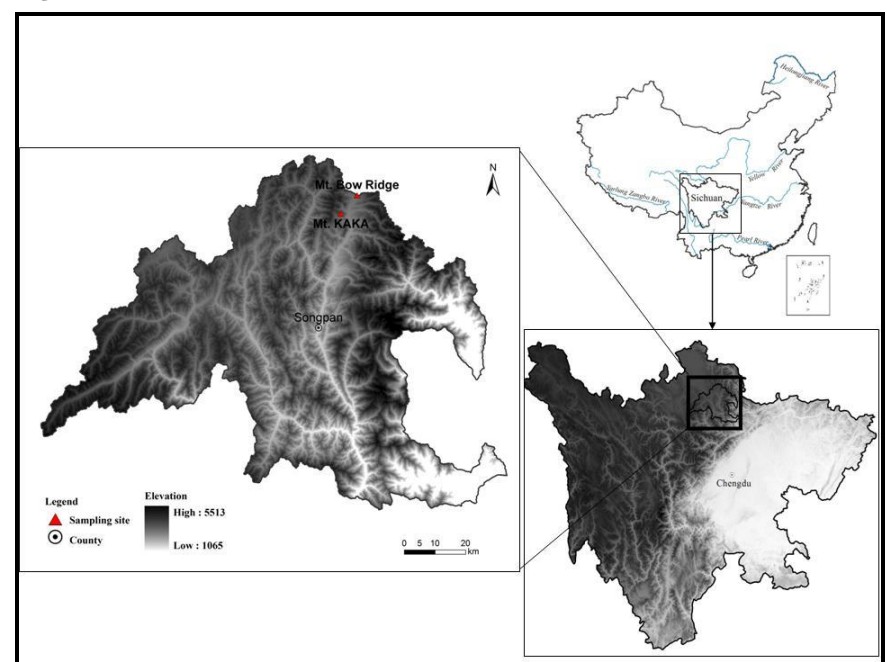


**Fig. 1** Location of the study sites.





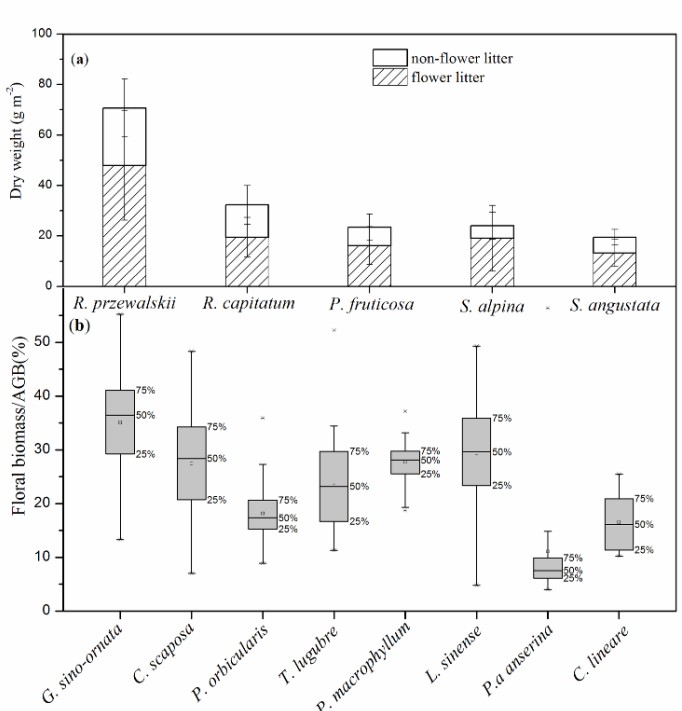


**Fig. 2** Production of flower litters and biomass allocation of representative dominant
species. (a) Production of flower litters and non-flower litters of shrubs
(phaenerophyte) per unit area (m$^2$); and (b) floral biomasses and their allocation in the
aboveground biomass.





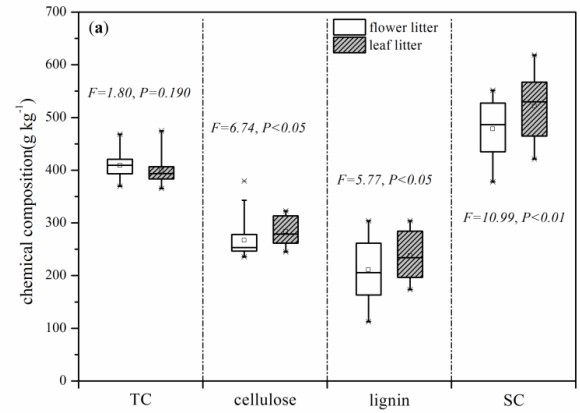


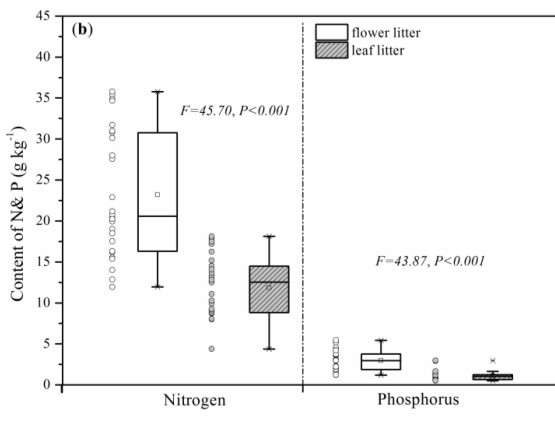


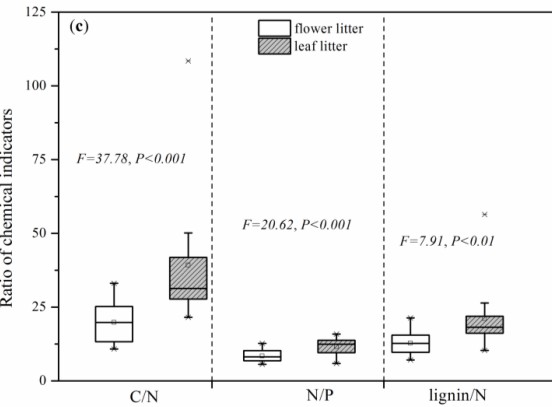


**Fig. 3** Chemical composition and their comparison between flower and leaf litters.














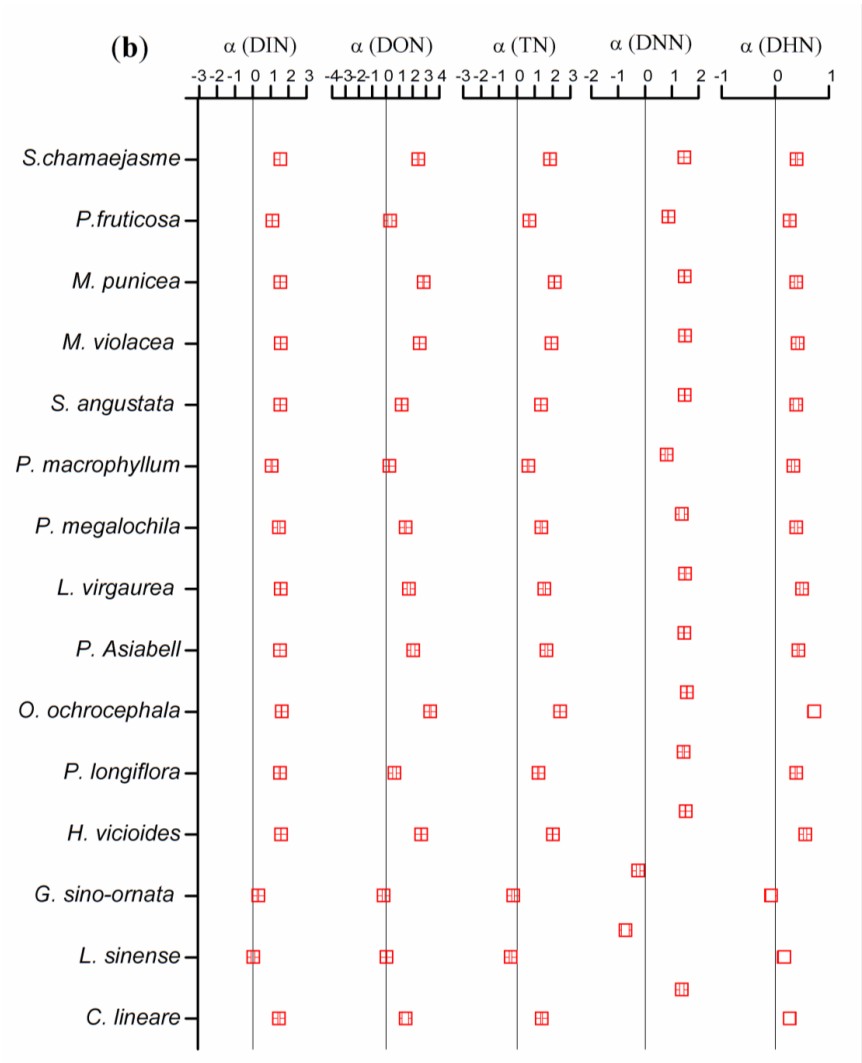


**Fig. 4** Variation in soil nitrogen pool after addition of flower litters.



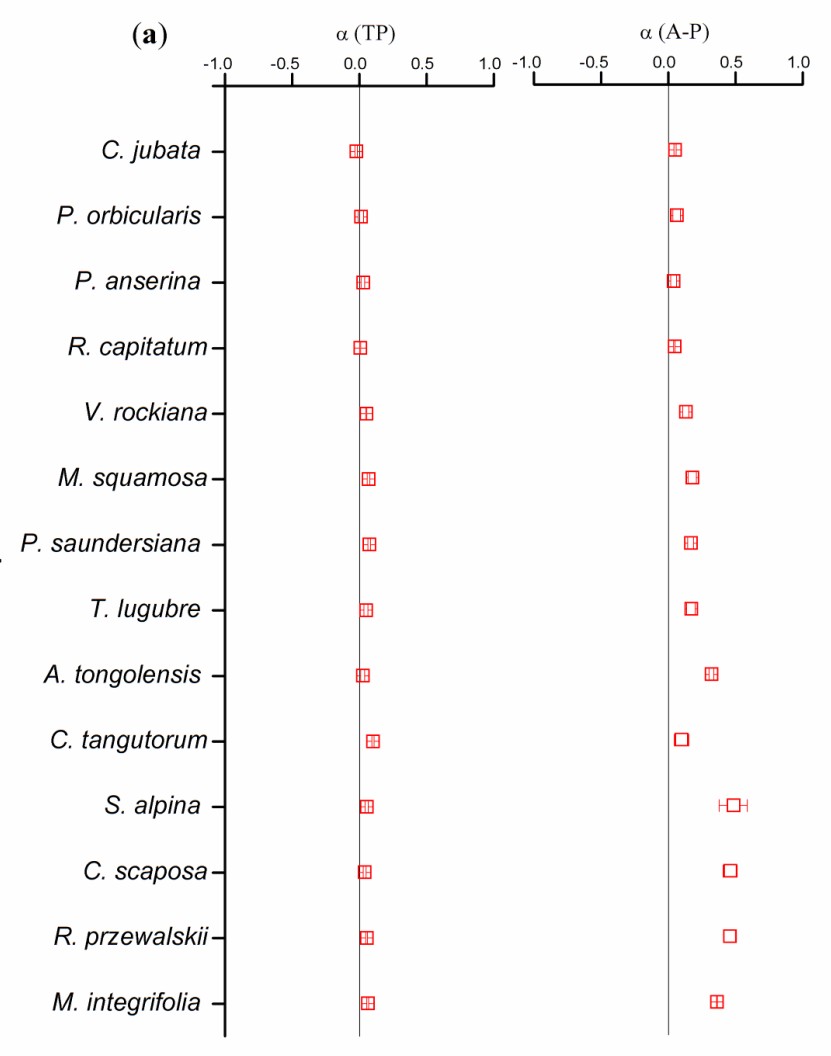






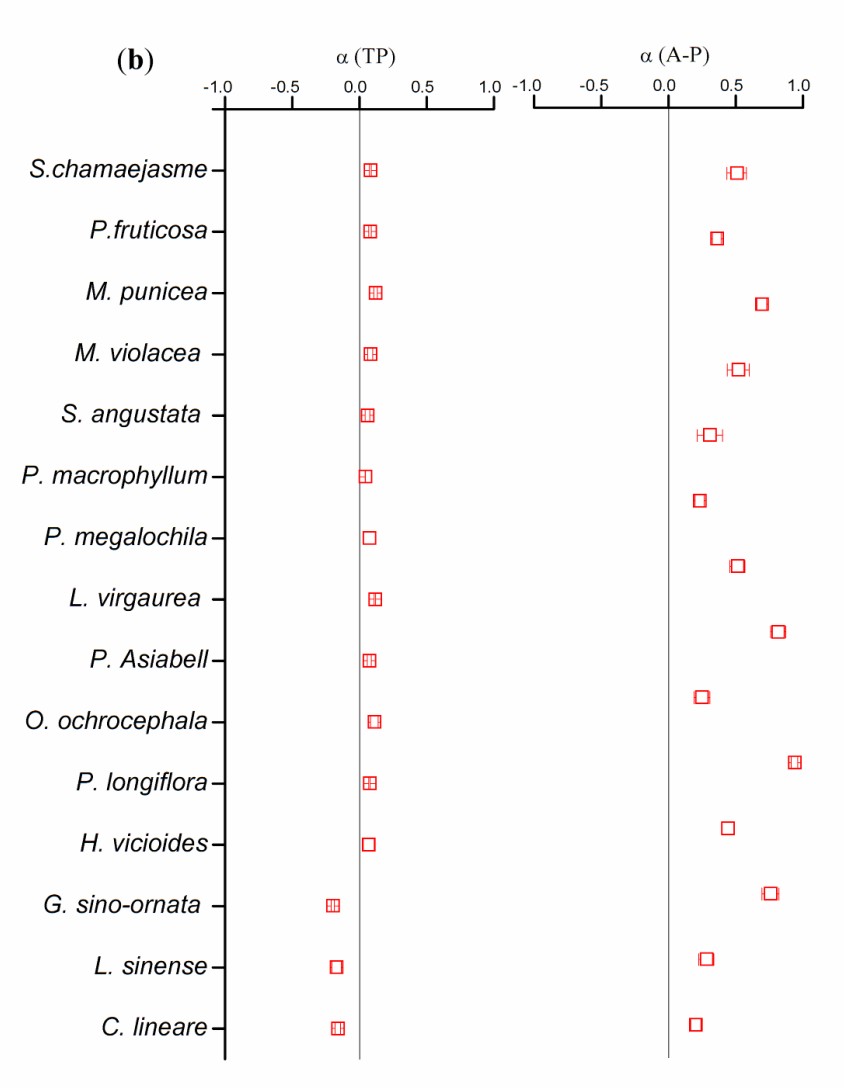


**Fig. 5** Variation of TP and A-P in flower litter added treatments.




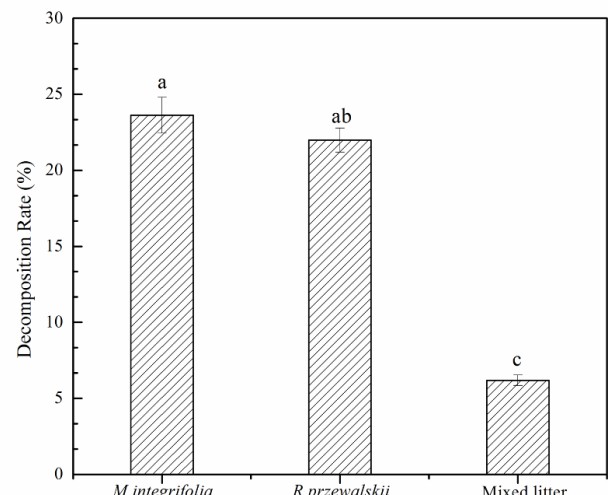


**Fig. 6** Percentage of decomposed dry mass of *M. integrifolia* and *R. przewalskii* in a
50-day litter-bag study. *Column* represents mean, and bar indicates Standard Error (n
= 8). Different lowercase letters indicate significant differences of decomposition rate
between litter materials.

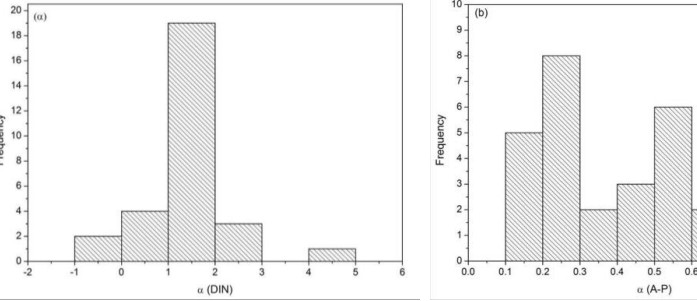


**Fig. 7** Variation in soil nutrition pool with flower litters addition. Histogram for α
values of DIN (a) and A-P (b) indicates the change between treatments and control.