# Peer review of "Flower litters of alpine plants affect soil nitrogen and phosphorus rapidly in the eastern Tibetan Plateau"

_Biogeosciences, 2016_

## Referee Comment (RC1) · Anonymous Referee #1 · 27 Apr 2016

Wang et al. dealt with the effects of flower litters on soil chemical properties. This is an interesting and important subject in nutrient cycling and plant growth. Authors gave good data supporting their hypothesis. However, employed methods are not clear, especially litter collection and chemical analyses of litters. Litter collection method is very important to assess the production of litter per unit area and researchers employ litter trap in general. However, authors did not give any specific methods, but said only "collect". Also, there is no method to analyse chemical properties of litter. For example, I could not find how authors measured total nitrogen and phosphorous in litter. Following should be considered. 1. Line 68: mineralization of soil organic matter and decomposition of plant residues have the same meaning. It is meaningless to devide

these two. 2. Line 152: Add "genus" before Kobresia and Carex and add spp. after Festuca, Gentiana, and Leontopodium to deliver exact meaning. 3. Line 164: Table 1 is not necessary. Delete it. 4. Line 164: Mixed litter- What is this litter composed of? It might be composed of flower and leaf litters. This is not clear. 5. Lines 184-187: It is not clear that collected samples were mixed through sieving or each sample was mixed through sieving. 6. Line 215: It is not necessary to use abb. of DHN and DNN for NH4+-N and NO3-N, respectively. 7. Table 1, 4 and 5 should go to the place after text which were mentioned. 8. Table 6 and explanation should go to results part, rather than Discussion part. 9. Line 259: delete per unit. 10. Table 2 and 3: DIN and DON are not necessary because these are deliberated from TN, DNN, and DHN and do not have any special meaning. 11. After delete DIN and DON, I suggest to combine Tables 2 and 4, Tables 3 and 5, and Figures 4 and 5. 12. Fig 4: add explanation of (a) and (b) in the Figure caption. 13. Fig 6: Where did ab of lower case letters come? To use ab, there should be b but there is no b. Check the statistical analysis.

———————————————————

---

## Author Comment (AC1) · 6 May 2016

Dear reviewer, All the authors greatly appreciated about your positive opinion and professional comments to our manuscript (bg-2016-68), which has improved its structure and readability. We are pleased to revise our manuscript in details according to your comments. Major comments about the methods, "However, employed methods are not clear, especially litter collection and chemical analyses of litters. Litter collection method is very important to assess the production of litter per unit area and researchers employ litter trap in general. However, authors did not give any specific methods, but said only "collect". Also, there is no method to analyse chemical properties of litter. For example, I could not find how authors measured total nitrogen and phosphorous in litter". Yes, we all agree that these items should have been added for their close relationship with interpreting the exact processes and following results in the context. The method of litter collection has been supplied in details, so have analysis methods of chemical properties. Litter collection, "4 litter traps were placed under the crown of each individual shrub in the study, which were processed and modified based on the litterfall monitoring protocol (Muller-Landau and Wright, 2010). The litter trap was composed of 1 cloth bag and 4 support legs. Window screen (with a mesh size of 0.8 mm) was used to seize the cloth bag. Its size was about 50 cm deep and 25 cm length of a side. 4 legs (made by 80 cm PVC pipe) were tied with cloth bag and frame. The frame of opening was made of iron wire with 3 mm diameter. After inserting it into the soil under the shrub's crown, the plant litter was collected twice per week, which was later sorted as flower litter and other types during the blooming period. Due to the small size of herbaceous individuals, flowers were just plucked at the end of flowering phase and their mass ratios to aboveground biomass was calculated. Freshly fallen leaves of different species were collected from the floor of the alpine meadow (i.e. mixed leaf litters, ca. 3950 m a.s.l.)." Chemical analyses, "The contents of C and N were determined by dry combustion with a CHNS auto-analyser system (Elementar Analysen systeme, Hanau, Germany) (Brodowski et al., 2006). The content of P was obtained colorimetrically by the chloro molybdophosphoric blue color method after wet digestion in a mixture of $HNO_3$, $H_2SO_4$, and $HClO_4$ solution (Institute of Soil Academia Sinica, 1978). Lignin and cellulose were estimated by the method described by Melillo et al (1989)." Muller-Landau, H.C. and Wright, S.J., 2010. Litterfall Monitoring Protocol. Melillo, J.M., Aber, J.D., Linkins, A.E., Ricca, A., Fry, B. and Nadelhoffer, K.J., 1989. Carbon and nitrogen dynamics along the decay continuum: Plant litter to soil organic matter. Plant and Soil 115, 189-198. Other minor comments 1. Line 68: mineralization of soil organic matter and decomposition of plant residues have the same meaning. It is meaningless to divide these two. Yes, we agree that there was no necessary to distinguish two terms since plant residues can be included in soil organic matter. In the revised version, we addressed the sentence to be "the plant residue is one principal component of soil organic matter, whose decomposition can supply available N to plants and microorganisms." 2. Line 152: Add "genus" before Kobresia and Carex and add spp. after Festuca, Gentiana, and Leontopodium to deliver exact meaning. Yes, we have done based on the review's suggestions. 3. Line 164: Table 1 is not necessary. Delete it. Ok, Table 1 has been deleted. 4. Line 164: Mixed litter- What is this litter composed of? It might be composed of flower and leaf litters. This is not clear. Thanks for your kind reminder. Here is mixed leaf litter, which was composed of dominant species' leaf litters in the alpine meadows. 5. Lines 184-187: It is not clear that collected samples were mixed through sieving or each sample was mixed through sieving. Yes, we have revised the relevant context-"After 50 days, each soil sample was collected from three points of each pot in the center and then mixed to avoid the boundary layer effect. Each soil sample from different PVC pots was mixed evenly by sieving through a 2 mm mesh respectively. The samples were stored and marked separately in an ice box prior to chemical determination." 6. Line 215: It is not necessary to use abb. of DHN and DNN for $NH_4^+$-N and $NO_3$-N, respectively. Thanks, $NH_4^+$-N and $NO_3$–N are just used in the updated version. 7. Table 1, 4 and 5 should go to the place after text which were mentioned. Yes, Table 4 and 5 have been edited to right place after combined with Table 2 and 3. 8. Table 6 and explanation should go to results part, rather than Discussion part. Yes, we did it after combine new figures and tables. Please kindly notice that it is Table 4 in revised version. Effects of flower litter addition on soil solution N pool and soil MBC and MBN Soil solution N pool has been improved noticeably from 31.46 mg g$-1$ to 47.35 mg g$-1$ in flower litter treatment compared with the control, particularly in fragment of $NO_3$–N, which has been greatly increased (from 30.93 mg g$-1$ to 46.8 mg g$-1$). (Table 4). In mixed leaf litter treatment, there were no obvious variations after litter decomposition, with 32.4 mg g-1 $NO_3$–N and 0.45 mg g-1 $NH_4^+$-N, respectively. There were notable differences of both MBC and MBN between different treatments. Litter addition not only increased soil microbial biomass C (102.05 mg kg$-1$, 68.08 mg kg$-1$, and 46.25 mg kg$-1$ for flower litter, mixed litter, and control, respectively) and MBN (73.02 mg kg$-1$, 69.29 mg kg$-1$, 67.13 mg kg$-1$ for flower litter, mixed litter, and control, respectively) but also their C:N ratios (1.40, 0.98, and 0.69 for flower litter, mixed litter, and control, respectively). Table 4 Comparing median value of soil solution pool and soil microbial biomass between litter addition treated (flower litter and mixed leaf litter) and control. Treatments Soil solution N pool (mg g-1) Soil microbial biomass (mg kg-1) NO3–N NH4+-N MBC MBN MBC/MBN Flower litter 46.8 0.55 102.05 73.02 1.40 Mixed leaf litter 32.4 0.45 68.08 69.29 0.98 Control 30.93 0.53 46.25 67.13 0.69

9. Line 259: delete per unit. Yes, done. 10. Table 2 and 3: DIN and DON are not necessary because these are deliberated from TN, DNN, and DHN do not have any special meaning. Yes, please see response of the 11th item. 11. After delete DIN and DON, I suggest to combine Tables 2 and 4, Tables 3 and 5, and Figures 4 and 5. Thanks for your professional recommendation, which improved the whole structure of this manuscript a lot. We have combined tables and figures for your further review. 12. Fig 4: add explanation of (a) and (b) in the Figure caption. Yes, we have added explanation of (a) and (b) in Fig 4. 13. Fig 6: Where did ab of lower case letters come? To use ab, there should be b but there is no b. Check the statistical analysis. Sorry for this unnecessary mistake. We have corrected in the revised version. We changed ab to be a and c to be b, respectively.

Thanks again for you time! Jinniu Wang, on behalf of all the authors of bg-2016-68

Please also note the supplement to this comment:
http://www.biogeosciences-discuss.net/bg-2016-68/bg-2016-68-AC1-supplement.pdf

**Effects of flower litter addition on soil solution N pool and soil MBC and MBN**

Soil solution N pool has been improved noticeably from 31.46 mg g$^{-1}$ to 47.35 mg g$^{-1}$ in flower litter treatment compared with the control, particularly in fragment of NO$_3^-$-N, which has been greatly increased (from 30.93 mg g$^{-1}$ to 46.8 mg g$^{-1}$). (**Table 4**). In mixed leaf litter treatment, there were no obvious variations after litter decomposition, with 32.4 mg g$^{-1}$ NO$_3^-$-N and 0.45 mg g$^{-1}$ NH$_4^+$-N, respectively. There were notable differences of both MBC and MBN between different treatments. Litter addition not only increased soil microbial biomass C (102.05 mg kg$^{-1}$, 68.08 mg kg$^{-1}$, 
[revised manuscript text omitted]

---

## Short Comment (SC1) · 11 May 2016

Wang et al. present a study about the flower litter and its roles in affecting the soil nitrogen and phosphorous, which is interesting and should attract scientific audience concerning the ecosystem resource cycle in alpine ecosystems. After reading the manuscript, several points should be addressed before acceptance to make the paper more sound and attractive . Line 42-44, the authors said the flower litters of phanerophyte plants were comparable with non-flower litters. To make it clear, the authors should point. The weight or something of litters are comparable. For the abbreviation, it should be mentioned for the first use, after that always use abbreviations. I suggest the authors to introduce why they also want to study P. Is N and P coupled in determin-

ing the storage and availability of soil resources? Line 159. Are the flower litters of 29 species collected in both sites. or just 14 for one site and 15 for another? Line 179. It seems you did not report the effect of leaf litter addition on decomposition. Line 205. You should make clear how many treatments in the decomposition experiment. To me, it seems there are three. Flower litter of two species and mixture of others. I guess the two species you mentioned should belong to early and later flowering groups, respectively.

Line 205. Can you make it clear how to determine the weight of litter after a period of time in the litter bag? Line 254. Can you compare the flower litter proportion to whole plant biomass in the two collecting groups or five life-form groups? Do the similar comparison for size of inflorescence?

Line 257-265. From the description in these lines, flower litter seems to account more than 60% if the non-flower litter represents biomass without flower. So please make it clear what the non-flower litter stands for, and make the difference between the non-flower litter and individual aboveground biomass

Line 277-278. I suggest put the F and P values after each indices. Line 293. The results you obtained based on the pooled data of all species. As you have measured the N and P of different species, can you present the results of interaction of species and different organs of plant on N and P. Line 313. As the result show no significant effect of interaction between flowering time and litter addition. If the nitrogen content and weight of flowers have no significant difference, TN and DON should have no significant difference. DIN and DNN might be the result of different priming effect of flower addition on soil mineralization rate.

Line 320. I suggest the authors put more emphasis on the DIN and DNN when investigating the effect of flower litter on soil nitrogen. As flowers have high N content, with and without litter addition should have significant difference even no experiment has been done because this relationship seem straightforward. However, for the DIN and

DNN, mineralization rate might contribute to the DIN and DNN.

Line 366. As you mentioned, in the MM section (Line 179). There should be four treatments, early flowering, later flowering, mixted leaf litter and control. I suppose you might make a typo. In line 179, it might be flower litter mixture.

Line 377. Make the flowering season specific.

Line 390. I suggest to add the information about the flower litter proportion to above-ground biomass in specific time and the whole growing season.

Line 422. Did I misunderstanding something? You discussed about the effect of C/N, lignin/N on leaf and flower on their decomposition, but you just reported the decomposition results of flower litter.

Line 438. I am not very familiar with the P cycling in the plant-soil. I guess the A-P comes from the soil and moves to the flower, after flower fall, it goes back to soil. I mean did the plant accelerate the weathering of minerals and contribute to the increased available P in the plant-soil. If not, it is just a redistribution of A-P in plant and soil at different times in the growing season and non-growing season. Line 501. I am not sure the requirement of the "Biogeoscience" to include a conclusion. It makes easier for reader to grab the major findings based on your discussion.

---

## Referee Comment (RC2) · Anonymous Referee #2 · 19 May 2016

The authors conducted a well-designed experiment to explore how flowering litter influences soil nitrogen and phosphorus status. I think this is an important topic that has not been well studied and deserves publication. However, I would also suggest the authors seek assistance with English grammar and translation where appropriate, as the impact of this paper is currently obscured.

The introduction of this paper is scattered and somewhat confusing. I think spending time re-organizing/framing this section will provide clarity for the results and discussion sections. Perhaps the authors could introduce the topic of flowering bodies and their higher litter quality (N content), then discuss how aboveground litter quality influences belowground biogeochemical cycling through microbial subsidies, and conclude with a

section discussing alpine ecosystems and evolved traits.

Lines 137-142 provide specific research questions that will be addressed by the authors. Currently they are a little unclear and seem to set up questions that are not directly tested. I would suggest refocusing on the major comparisons being made- is flower litter of higher quality than leaf litter? do these traits facilitate faster decomposition? does the time of litter fall influence ecosystem productivity?

The methods/materials section is generally clear. However, the authors do not provide details regarding their litter collection method (makes question 1 difficult to assess). The number of replicates and control treatments are sound. In my personal opinion I think it is important the field moves beyond litter bag experiments and mass loss. Litter bags exclude fauna and litter fragmentation, which contribute greatly to litter decomposition. While the authors used mesh bags with two layers of differing mesh size, the smaller mesh actually surrounding the litter still excludes faunal decomposers and minimizes biophysical perturbation. Since the study is focused on nutrient cycling more than soil organic matter mass loss/formation I think the litter bag approach is okay, but in the future it would be good for us to move beyond these techniques.

A-P is never defined in the manuscript. Correction factors for microbial biomass C and N are commonly employed, but are highly specific to soil mineraology/sorption. Direct testing of recovery efficiency at a particular site should be assessed before a correction factor is applied.

Line 357: clarify which species were used to compare decomposition rates between flower and mixed litters.

Line 397: lignin/N and C/N ratios are commonly accepted as good indicators of decomposition rates under short time frames, but there is little evidence lignin is preferentially preserved in soils, compared with bulk soil, over long-time periods (Cotrufo et al., 2015 &2015, Mikutta et al., 2005 Kleber et al., 2007 etc).

Line 467: microbial community composition was not directly tested (no sequencing/PLFA analysis etc) so it cannot be concluded that flowering litter increases nutrient status and therefore changes microbial assembly. There is support that MBC and MBN pools increase, but that could be due to faster turnover or growth, not necessarily to a change in species composition.

Line 499: the impact of this paper is significant and should be re-stated clearly in the conclusion.

Comments regarding tables: Make sure to clearly define variables tested in each caption. Table 1: it is not clear how species dominance is assessed (Y/N). Line 303, 324, 329 etc.: the authors are assessing N pools, not fragments. Table 3: DNN/DHN are not necessary; although defined as such in the text, $NO_3^-$ and $NH_4^+$ are clearer. Table 4/5: Define TP and A-P: $\alpha$ values of total phosphorus (TP)... Table 6: Mean values (not comparison medium values)

Comments regarding figures: Figure 1: It is very difficult to identify where the sampling sites are on the map because the elevation shading is so dark (either increase shading transparency or make text and symbols larger) Figure 2: Include the mean (n=X). Figure 3: Explicitly state the statistical analysis used (are the bars 95% confidence intervals/SE, or quantiles)? If the whiskers represent SE it seems impossible that the flower litter vs. leaf litter means are significantly different from each other. What are the values (mean, n=X)? Figures 4 & 5: Define the variables in the figure caption (dissolved inorganic nitrogen (DIN), dissolved organic nitrogen (DON), etc). mean, n=X. What do the boxes represent? What does deviation from the 0 line signify (significantly different at what level)? Figure 6: letters indicate significant differences (at what level, p=0.05)?

Comments regarding cited literature Overall, the authors seem well read on these topics. There are some citations that do not seem relevant to the paper the way it is currently written (for example, findings from tropical, agricultural, and Arctic sites). I think these findings are important because they show flowering litter quality influences

soil nutrient status across ecotones, but this needs to be made explicitly clear. The authors also touch on soil organic matter formation/stabilization processes. There is a highly relevant body of literature that could be incorporated to strengthen these points (Kogel-Knaber, Sollins, Cotrfuo, Kleber, etc).

Overall, I think the quality of this study is excellent. I would highly suggest a thorough language editing review is taken before the paper is published. Currently it is a little difficult to read and I think this obscures the interesting science.

---

## Author Comment (AC2) · 19 May 2016

Dear Peng,

Thanks for your interest in our study published in BGD (bg-2016-68). Some items you mentioned in your comments are quite reasonable, which were also raised in the previous referee's comments. We would like to answer your concerned points one by one (Q, underline, and A, plain).

Line 42-44, the authors said the flower litters of phanerophyte plants were comparable with non-flower litters. To make it clear, the authors should point. The weight or something of litters are comparable. For the abbreviation, it should be mentioned for the first use, after that always use abbreviations. I suggest the authors to introduce why they also want to study P. Is N and P coupled in determining the storage and availability of soil resources?

Yes, it is dry weight. Then, "for phanerophyte plants, it was comparable of dry weight between flower litters and non-flower litters". For the abbreviation, it will be revised as you suggested. In addition to the reason to study P had been mentioned in Line 109, here are some other points; 1) according to the previous study, large quantities of tree pollen can be produced over a relatively short period in early summer (Doskey & Ugoagwu, 1989) in many temperate forest ecosystems; 2) they can play an important atmospheric source of macronutrients in terrestrial and aquatic communities for their high nutrient concentrations and high decomposition rate (Stark, 1972; Doskey & Ugoagwu, 1989); 3) in boreal forests, pollen may also be an important role in adding nutrients and promoting decomposition (Lee et al., 1996). The last point, N and P must be coupled in determining the storage and availability of soil resources since it is one core content of ecological stoichiometry.

Line 159. Are the flower litters of 29 species collected in both sites. or just 14 for one site and 15 for another?

Yes, they were collected from both sites.

Line 179. It seems you did not report the effect of leaf litter addition on decomposition.

It had been comprehensively presented in table 6 about the effect of leaf litter addition on decomposition. As we explained about the experiment design, there was one treatment about mixed leaf litter addition from alpine meadow during the blooming period.

Line 205. You should make clear how many treatments in the decomposition experiment. To me, it seems there are three. Flower litter of two species and mixture of others. I guess the two species you mentioned should belong to early and later flowering groups, respectively.

There are three treatments, which consisted of flower litter of *R. przewalskii*, flower litters of *M. integrifolia*, and mixed leaf litter. All of the above which were put into litter bags. Two species mentioned as representative species were from dominant shrub species for their wide distribution and massive flower litter production.

Line 205. Can you make it clear how to determine the weight of litter after a period of time in the litter bag?

Yes, we would like to modify and add more information in details. Firstly, remove the debris or mud outside the litter bags carefully, then litter was taken outside and sank into small water basin for short period of time (30 minute), which would go through 0.5 mm mesh filter to sort out clay and litter. Lastly, litters were dried at 60℃ in an oven for 48 hours and measured the weight on the balance (accuracy 0.001 g).

Line 254. Can you compare the flower litter proportion to whole plant biomass in the two collecting groups or five life-form groups? Do the similar comparison for size of inflorescence?

Because there are not only grasses (herbaceous plants) but some shrubs (phanerophyte plants), we could not calculate and compare the proportion of flower litter to the whole plant biomass unless the whole shrub individual were dug out, which is not practical and available. Moreover, only the whole aboveground part of grasses will wither with some parts fallen into soil for decomposition. Thus, we compared proportion of flower litter differently according to phanerophyte or herbaceous plant since the approaches of litter collection were different (litter trap or manually pluck), respectively. The information of size of inflorescence had been deleted according to the previous referee's comments. We also thought that point over for its reasonability because plants may produce smaller size but lots of flowers, and pollen mass may differ from certain species. Hope we understood your point properly.

Line 257-265. From the description in these lines, flower litter seems to account more than 60% if the non-flower litter represents biomass without flower. So please make it clear what the non-flower litter stands for, and make the difference between the non-flower litter and individual aboveground biomass

Yes, similar as the previous one. When we compared flower litter with non-flower litter, those species were phanerophyte plants but not herbaceous plants, whose litters can be collected by litter trap (the process had been added in details). Here non-flower litters represented litters in the litter traps excluding flower litters (i.e. leaf and twig). This point had been described "the flower litters of phenerophyte plants, whose flower litters …"

Line 277-278. I suggest put the F and P values after each indices. Line 293. The results you obtained based on the pooled data of all species. As you have measured the N and P of different species, can you present the results of interaction of species and different organs of plant on N and P.

Yes, we will do that. Previously, we just did pooled data of all species since the detailed information had been illustrated in Fig. 4 and Fig. 5.

Line 313. As the result shows no significant effect of interaction between flowering time and litter addition on TN and DON. If the nitrogen content and weight of flowers have no significant difference, TN and DON should have no significant difference. DIN and DNN might be the result of different priming effect of flower addition on soil mineralization rate.

We agree that "priming effect" of flower addition had effects on DIN and DNN (also see Line 479). However, in this study, TN and DON had significant differences, which can be understood if we take the species-specific size and pollen production into account (e.g. shrub, *Rhododendron przewalskii* versus herbaceous plant, *Primula orbicularis*).

Line 320. I suggest the authors put more emphasis on the DIN and DNN when investigating the effect of flower litter on soil nitrogen. As flowers have high N content, with and without litter addition should have significant difference even no experiment has been done because this relationship seem straightforward. However, for the DIN and DNN, mineralization rate might contribute to the DIN and DNN.

Yes, agree. It had been also suggested by the previous referee. We deleted DON and DIN, and combined TP and A-P together. Also, mineralization rate should be emphasized more.

Line 366. As you mentioned, in the MM section (Line 179). There should be four treatments, early flowering, later flowering, mixed leaf litter and control. I suppose you might make a typo. In line 179, it might be flower litter mixture.

From a certain extent, we agree to your point. If we address experimental design according to flowering time but not exactly different species, then there would be four treatments. However, we were considering different species from two flowering period (earlier and later), so that there should be 33 treatments (31 flower litters from different species, 1 mixed leaf litter, and 1 control, in total 33 treatments).

Line 377. Make the flowering season specific.

Ok, we will add some points about the specific flowering seasons.

Line 390. I suggest to add the information about the flower litter proportion to aboveground biomass in specific time and the whole growing season.

Please kindly see the reply to Line 254 and Line 257-265. For those herbaceous plants, the proportion of their flower litter to aboveground biomass is available unlike other shrubs.

Line 422. Did I misunderstanding something? You discussed about the effect of C/N, lignin/N on leaf and flower on their decomposition, but you just reported the decomposition results of flower litter.

Yes. Because we could not collect enough amount of leaf litters of some species during blooming period. However, some leaf litters were used for determination of chemical composition, we just compared chemical properties of dominant species between flower litter and leaf litter. In the decomposition experiment, mixed leaf litter of herbaceous plants in an alpine meadow was adopted instead of all species for the similar reason. Moreover, it can be up-front that those herbaceous plants produce more decomposable leaves for less wax and thinner cuticles than those shrubs in alpine ecosystems, at least for those species in this study.

Line 438. I am not very familiar with the P cycling in the plant-soil. I guess the A-P comes from the soil and moves to the flower, after flower fall, it goes back to soil. I mean did the plant accelerate the weathering of minerals and contribute to the increased available P in the plant-soil. If not, it is just a redistribution of A-P in plant and soil at different times in the growing season and non-growing season.

From our understanding, A-P comes from soil and will be absorbed by plant root systems. Then, P will be prioritized to reproduction organ along with the allocation of accumulated carbohydrates, which is an adaptation strategy of alpine plants in a hush environment (i.e. alpine ecosystem). It is not easy to distinguish weathering of minerals accelerated by plant or redistribution of A-P in plant and soil because the decomposition of plant litters is a long process for several growing seasons and non-growing seasons.

Line 501. I am not sure the requirement of the "Biogeoscience" to include a conclusion. It makes easier for reader to grab the major findings based on your discussion.

Thanks for you kind reminder. We will wait for editor's recommendation.

Thanks again for your time!

Jinniu Wang on behalf of all authors

---

## Author Comment (AC3) · 10 Jun 2016

Dear Reviewer,

Many thanks for your kind comments on our manuscript. We also greatly appreciate the associate editor (Dr. Richard) for inviting some experts who know quite well about this topic. Thus, there is no doubt that we are very happy to make a revision regarding your constructive statements. For your convenience, we attached English editing certificate, and reply to referee point-to-point and revised version of MS as supplement files (zip).

Thanks again for your time!

Jinniu Wang on behalf of all Co-authors

[Figure]

Please also note the supplement to this comment:
http://www.biogeosciences-discuss.net/bg-2016-68/bg-2016-68-AC3-supplement.zip
* * *
EnPapers.Com  CERTIFICATE OF ENGLISH EDITING

To whom it may concern:

This memo is to certify that the paper titled Flower Litters of Alpine Plants Rapidly Affect Soil Nitrogen and Phosphorus in the Eastern Tibetan Plateau has been edited for language by EnPapers, a company dedicated to helping international researchers publish their findings in the best English language journals possible.

Our International paper editing service is performed by a subject expert editor and approved by two senior editors. All our editors are native English speakers.

The certificate is being issued upon the request of the client. If you have any questions, please contact papers@enpapers.com

---

## Author Response (AR1)

Dear Editor Dr. Richard, and two anonymous referees,

We appreciate all your comments greatly on our previous draft entitled "Flower litters of alpine plants affect soil nitrogen and phosphorus rapidly in the eastern Tibetan Plateau" (bg-2016-68). The revised version has been completed based on your favorable suggestions item by item. Besides, all text after revision have been yellow highlighted in the new version for your consideration, except the introduction part since it has been totally re-organized according to Referee #2's recommendation.

**Reply to referee #1**

**Major comments about the methods,**

"**However, employed methods are not clear, especially litter collection and chemical analyses of litters. Litter collection method is very important to assess the production of litter per unit area and researchers employ litter trap in general. However, authors did not give any specific methods, but said only "collect". Also, there is no method to analyse chemical properties of litter. For example, I could not find how authors measured total nitrogen and phosphorous in litter**".

Thanks. Yes, these details should have been added for better interpreting the exact processes and following results in the context. We have supplied the method of litter collection and analysis methods of chemical properties. Besides, total phosphorus (TP) and available phosphorous (A-P) have been clarified.

**See Line 155-168**. **Litter collection**, "In the study, 4 litter traps were placed under the crown of each individual shrub in different communities (5–8 individuals were chosen for the placement of litter traps), which were processed and modified based on the litterfall monitoring protocol (Muller-Landau and Wright, 2010). The litter trap was composed of 1 cloth bag and 4 support legs. Window screen (with a mesh size of 0.8 mm) was used to seize the cloth bag. Its size was about 50 cm deep and 25 cm long. Four legs (made with 80 cm PVC pipe) were tied with a cloth bag and frame. The frame of the opening was made of iron wire with 3 mm diameter. After inserting it into the soil under the shrub's crown, the plant litter was collected twice per week, which was later sorted as flower litter and other types during the blooming period. Given the small size of herbaceous individuals, flowers were plucked at the end of the flowering phase, and their mass ratios to aboveground biomass were calculated. Freshly fallen leaves of different species were collected from the floor of the alpine meadow (i.e., mixed leaf litters, ca. 3950 m a.s.l.).

**See Line 250-258**. **Definition of TP and A-P**: Total phosphorus (TP) consists of phosphorus mineral and organic phosphorous compound in the soil, which can be converted into the dissolved orthophosphate. Available phosphorous (A-P) is the fragments in soil that can be absorbed by plants, which consist of water-soluble phosphorus, some adsorbed phosphorus, organic phosphorus, and precipitated phosphorus in certain soil types. Chemically, A-P is defined as the phosphorus and phosphate in soil solution that can be isotope exchanged with $^{32}P$ or can be easily extracted by some chemical reagents. TP and A-P in soils were estimated by extraction with 0.5 M sodium hydroxide sodium carbonate solution (Dalal, 1973).

**See Line 262-268**. **Chemical analyses**, "For plant samples, the contents of C and N were determined by dry combustion with a CHNS auto-analyzer system (Elementar Analysen Systeme, Hanau, Germany) (Brodowski et al., 2006). The content of P was obtained colorimetrically by the chloro molybdophosphoric blue color method after wet digestion in a mixture of $HNO_3$, $H_2SO_4$, and $HClO_4$ solution (Institute of Soil Academia Sinica, 1978). Lignin and cellulose were estimated by the method described by Melillo et al. (1989).

References:

Muller-Landau, H.C. and Wright, S.J., 2010. Litterfall Monitoring Protocol.

Melillo, J.M., Aber, J.D., Linkins, A.E., Ricca, A., Fry, B. and Nadelhoffer, K.J., 1989. Carbon and nitrogen dynamics along the decay continuum: Plant litter to soil organic matter. Plant and Soil 115, 189-198.

**Other minor comments**

**1. Line 68: mineralization of soil organic matter and decomposition of plant residues have the same meaning. It is meaningless to divide these two**.

Yes, we agree that there was no need to distinguish two terms since plant residues can be included in soil organic matter. In the revised version, we addressed the sentence to be "the plant residue is one principal component of soil organic matter, whose decomposition can supply available N to plants and microorganisms." **See Line 89-91**.

**2. Line 152: Add "genus" before *Kobresia* and *Carex* and add spp. after *Festuca*, *Gentiana*, and *Leontopodium* to deliver exact meaning.**

Thanks, we added. **See Line 144.**

**3. Line 164: Table 1 is not necessary. Delete it.**

Sorry, after discussed with each other and also referred to the comments from referee #2, we would like to keep table 1 for better presenting results. Besides, we revised description of relevant text.

**4. Line 164: Mixed litter- What is this litter composed of? It might be composed of flower and leaf litters. This is not clear.**

Thanks for your kind reminder. It is mixed leaf litter, which was composed of dominant species' leaf litters in the alpine meadows.

**5. Lines 184-187: It is not clear that collected samples were mixed through sieving or each sample was mixed through sieving.**

Thanks, it has been revised-"After 50 days, each soil sample was collected from three points of each pot in the center and then mixed to avoid the boundary layer effect. Each soil sample from different PVC pots was mixed evenly by sieving through a 2 mm mesh respectively. The samples were stored and marked separately in an ice box prior to chemical determination." **See Line 207-211.**

**6. Line 215: It is not necessary to use abb. of DHN and DNN for NH4+-N and NO3-N, respectively.**

Thanks, $NH_4^+$-N and $NO_3^-$-N are just used in the latest version.

**7. Table 1, 4 and 5 should go to the place after text which were mentioned.**

Yes, Table 4 and 5 have been edited to right place after combined with Table 2 and 3.

**8. Table 6 and explanation should go to results part, rather than Discussion part.**

Ok, please kindly notice that it is **Table 4** in revised version after pooled new figures and tables. **See Line 373-387.**

**Effects of flower litter addition on soil solution N pool and soil MBC and MBN**

Soil solution N pool has been improved noticeably from 31.46 mg $g^{-1}$ to 47.35 mg $g^{-1}$ in flower litter treatment compared with the control, particularly in fragment of $NO_3^-$-N, which has been greatly increased (from 30.93 mg $g^{-1}$ to 46.8 mg $g^{-1}$). (**Table 4**). In mixed leaf litter treatment, there were no obvious variations after litter decomposition, with 32.4 mg $g^{-1}$ $NO_3^-$-N and 0.45 mg $g^{-1}$ $NH_4^+$-N, respectively. There were notable differences of both MBC and MBN between different treatments. Litter addition not only increased soil microbial biomass C (102.05 mg $kg^{-1}$, 68.08 mg $kg^{-1}$, and 46.25 mg $kg^{-1}$

for flower litter, mixed litter, and control, respectively) and MBN (73.02 mg kg$^{-1}$, 69.29 mg kg$^{-1}$, 67.13 mg kg$^{-1}$ for flower litter, mixed litter, and control, respectively) but also their C:N ratios (1.40, 0.98, and 0.69 for flower litter, mixed litter, and control, respectively).

**Table 4** Comparing median value of soil solution pool and soil microbial biomass between litter addition treated (flower litter and mixed leaf litter) and control.

| Treatments | Soil solution N pool (mg g$^{-1}$) | | Soil microbial biomass (mg kg$^{-1}$) | | |
|---|---|---|---|---|---|
| | NO$_3^-$-N | NH$_4^+$-N | MBC | MBN | MBC/MBN |
| Flower litter | 46.8 | 0.55 | 102.05 | 73.02 | 1.40 |
| Mixed leaf litter | 32.4 | 0.45 | 68.08 | 69.29 | 0.98 |
| Control | 30.93 | 0.53 | 46.25 | 67.13 | 0.69 |

**9. Line 259: delete per unit.**

Yes, done.

**10. Table 2 and 3: DIN and DON are not necessary because these are deliberated from TN, DNN, and DHN do not have any special meaning.**

Yes, please notice the response of the item 11 above.

**11. After delete DIN and DON, I suggest to combine Tables 2 and 4, Tables 3 and 5, and Figures 4 and 5.**

Thanks for your professional recommendation, which greatly improved the manuscript's whole structure. Tables and figures have been combined for your further review. **See Line 325-371, and Line 699-702.**

**12. Fig 4: add explanation of (a) and (b) in the Figure caption.**

Yes, we have added more detail explanation of (a) and (b) in Fig 4. **See Line 703-707.**

**13. Fig 6: Where did ab of lower case letters come? To use ab, there should be b but there is no b. Check the statistical analysis.**

Sorry for this unnecessary mistake. It has been corrected in the revised version (kindly refer to the below, which is Fig. 5 in the revised version).

[Figure]

**Reply to referee #2**

**Major comments**

"**However, I would also suggest the authors seek assistance with English grammar and translation where appropriate, as the impact of this paper is currently obscured**."

Ok, we agree that it is necessary to seek assistance from native English speakers. After revision of the whole structure in details, this paper has been sending to professional polish company with attached certificate for your consideration.

"**The introduction of this paper is scattered and somewhat confusing. I think spending time re-organizing/framing this section will provide clarity for the results and discussion sections. Perhaps the authors could introduce the topic of flowering bodies and their higher litter quality (N content), then discuss how aboveground litter quality influences belowground biogeochemical cycling through microbial subsidies, and conclude with a section discussing alpine ecosystems and evolved traits**."

Thanks, very good recommendation. We revised introduction part according to your kind suggestions for better addressing scientific questions, results, and discussion sections. Please see **Line 66-133**.

**Other comments**

"**Lines 137-142 provide specific research questions that will be addressed by the authors. Currently they are a little unclear and seem to set up questions that are not directly tested. I would suggest refocusing on the major comparisons being made- is flower litter of higher quality than leaf litter? do**

**these traits facilitate faster decomposition? does the time of litter fall influence ecosystem productivity?”**

Yes, the draft will be definitely improved if we focus on the major comparison. Thus, after supplying method in details of litter collection, we revised three questions as follows: 1) Should flower litter be considered in the alpine ecosystem's biogeochemical cycles for their relatively innegligible biomass production and/or allocation? 2) Does flower litter of higher quality and with unique traits have faster decomposition than leaf litter? 3) Does the time of litter fall influence soil available nutrients and soil microbial productivity of alpine meadow ecosystem?  See **Line 128-133**.

“**The methods/materials section is generally clear. However, the authors do not provide details regarding their litter collection method (makes question 1 difficult to assess). The number of replicates and control treatments are sound. In my personal opinion I think it is important the field moves beyond litter bag experiments and mass loss. Litter bags exclude fauna and litter fragmentation, which contribute greatly to litter decomposition. While the authors used mesh bags with two layers of differing mesh size, the smaller mesh actually surrounding the litter still excludes faunal decomposers and minimizes biophysical perturbation. Since the study is focused on nutrient cycling more than soil organic matter mass loss/formation I think the litter bag approach is okay, but in the future it would be good for us to move beyond these techniques**.”

Thanks, we suppose this is similar as the referee #1 raised. Litter collection procedure has been added in the revised version. Yes, you are quite right. The smaller mesh affected faunal decomposers and minimizes biophysical perturbation of litters to some extent. We did similar kind of comparison with quantification method between different meth sizes. If chose smaller size, tiny size litter can be hold in the litter bag better, but it may exclude some soil fauna. If chose bigger size, tiny litters might be easily dropped in the procedure. We do agree reviewer's constructive recommendation that in the future it would be great to move beyond these techniques to closely simulate real statues. For example, dual-labelled litter has been applied in recent decomposition studies.

**Litter collection method**: In the study, 4 litter traps were placed under the crown of each individual shrub in different communities (5–8 individuals were chosen for the placement of litter traps), which were processed and modified based on the litterfall monitoring protocol (Muller-Landau and Wright, 2010). The litter trap was composed of 1 cloth bag and 4 support legs. Window screen (with a mesh size of 0.8 mm) was used to seize the cloth bag. Its size was about 50 cm deep and 25 cm long. Four legs

(made with 80 cm PVC pipe) were tied with a cloth bag and frame. The frame of the opening was made of iron wire with 3 mm diameter. After inserting it into the soil under the shrub's crown, the plant litter was collected twice per week, which was later sorted as flower litter and other types during the blooming period. Given the small size of herbaceous individuals, flowers were plucked at the end of the flowering phase, and their mass ratios to aboveground biomass were calculated. Freshly fallen leaves of different species were collected from the floor of the alpine meadow (i.e., mixed leaf litters, ca. 3950 m a.s.l.).

**Determine the weight of litter after decomposition**: Firstly, the debris or mud was remove outside the litter bags carefully, then litter was taken outside and sank into small water basin for short period of time, which would go through 0.5 mm mesh filter to sort out clay and litter. Lastly, litters were dried at 60℃ in an oven for 48 hours and measured the weight on the balance (accuracy 0.001 g). See **Line 230-235**.

"**A-P is never defined in the manuscript. Correction factors for microbial biomass C and N are commonly employed, but are highly specific to soil mineraology/sorption. Direct testing of recovery efficiency at a particular site should be assessed before a correction factor is applied**."

Sorry for missing definition of A-P. It has been supplied in the methods section. Total phosphorus (TP) consists of phosphorus mineral and organic phosphorous compound in the soil which can be converted into the dissolved orthophosphate. Available phosphorous (A-P) is the fragments in soil can be absorbed by plants, which consist of water solvable phosphorus, some adsorbed phosphorus, and organic phosphorus, even including precipitated phosphorus in certain soil types. Chemically, A-P is defined that phosphorus and phosphate in soil solution can be isotope exchanged with $^{32}$P or can be easily extracted by some chemical reagents. see **Line 250-258**.

We agree with your professional point that correction factors for microbial biomass C and N are highly specific at different soil sampling sites or soil types. This issue has been discussed with the lab staff in our institute, who also referred the same opinion about methodology of MBC and MBN given it was processed by general international method and relevant correction factor. However, as referee also mentioned that this paper is not specially focused on the microorganism scope, this flaw will not make a decisive change to the results and conclusions, for the same soil type and also fully mixed through sieve before litter addition. Surely, we will pay more attention to site-specific correction factors in the future's research. Many thanks for your helpful suggestions.

"**Line 357: clarify which species were used to compare decomposition rates between flower and mixed litters**."

Yes, we have clarified. *R. przewalskii* and *M. integrifolia* are two typical plant species widely distributed and easily collected. Both species were assessed to compare their decomposition rates of flower litter and mixed leave litter.

"**Line 397: lignin/N and C/N ratios are commonly accepted as good indicators of decomposition rates under short time frames, but there is little evidence lignin is preferentially preserved in soils, compared with bulk soil, over long-time periods (Cotrufo et al., 2015, Mikutta et al., 2005, Kleber et al., 2007)**."

Yes, we have revised previous statement as follows: "Generally, tissues with high lignin, polyphenol, and wax contents and higher lignin/N and C/N ratios exhibit slow decomposition. Lignin/N and C/N ratios are commonly accepted as good indicators of decomposition rates under short time frames, but there is little conclusive evidence that lignin is preferentially preserved in soils, compared with bulk soil, over long-time periods (Melillo et al., 1982; Mikutta et al., 2005; Kleber et al., 2007; Cotrufo et al., 2015). Moreover, lignin plays dual role in plant litter decomposition if taken photochemical mineralization these abiotic decomposition into account (Austin and Ballaré, 2010)". According to the literature from Austin and Ballaré (PNAS, 2010), it was said that biotic decomposition in mesic ecosystems is generally negatively correlated with the concentration of lignin, which is a typically recalcitrant material that is resistant to microbial decomposition. However, for its dual role in plant litter decomposition, lignin is quite complicated if we take photochemical mineralization of carbon into account. See **Line 429-437**.

"**Line 467: microbial community composition was not directly tested (no sequencing/PLFA analysis etc) so it cannot be concluded that flowering litter increases nutrient status and therefore changes microbial assembly. There is support that MBC and MBN pools increase, but that could be due to faster turnover or growth, not necessarily to a change in species composition**."

Yes, this not quite convincing deduction has been revised since there was no direct evidence. We have been processing another study this year, which is aiming to compare the microbial community composition (sequencing) in soil after flower litter addition (two dominate shrubs- *Rhododendron capitatum* and *Rhododendron przewalskii*), which has been conducted both in the field and incubator.

We have revised the relevant statement: "Flower litter contains more than twice MBC (increased from 46.25 to 102.05), and both MBC and MBN pools increased potentially after flower litter addition.

Therefore, microbial functional groups might be changed for nutrient supplement from litters, or could also be due to their faster turnover or growth, which need more evidences in the further study by directly testing of soil microbial community composition. See **Line 501-506**.

"**Line 499: the impact of this paper is significant and should be re-stated clearly in the conclusion**."

Thanks for your compliment. We have re-stated the significance of this paper in the last paragraph. See **Line 535-546**.

**Comments regarding tables**

"**Make sure to clearly define variables tested in each caption**."

**Table 1: it is not clear how species dominance is assessed (Y/N).**

We have added in the method part about how to assess dominance of species for both shrub and herbaceous species in the study sites. Target species were firstly decided by visual observation. For herbaceous species, their dominances were determined using quadrat methods. Each quadrat (1 m × 1 m) was spaced at least 2 m apart from each other along the transect for recording community composition (totaling 10 quadrats along one transect, and three transects at each site). Weighted means of frequency and biomass of target species were sorted and used to assess their dominances. For shrubs, line-point intercept method was conducted to calculate targeted species' frequency, height, and cover, which represented by "hit" (3 transects at each site, use 20 m rope with ca. 1 cm diameter or measuring tape), whose weighted means of were sorted to determine dominant species (Herrick et al., 2005). We also consulted expert who already has prior knowledge or researches about the dominant species at the selected sites. See **Line 170-181**.

Reference: Herrick, J.E., Van Zee, J.W., Havstad, K.M., Burkett, L.M. and Whitford, W.G., 2005. Monitoring manual for grassland, shrubland and savanna ecosystems. Volume I: Quick Start. Volume II: Design, supplementary methods and interpretation. USDA-ARS Jornada Experimental Range.

**Line 303, 324, 329 etc.: the authors are assessing N pools, not fragments.**

Thanks for this precise comment. We combined Fig. 4 and Fig. 5 together and deleted DIN and DON. Besides, the relevant content has been revised regarding the modified figures, in particular, focused more on N pools.

**Table 3: DNN/DHN are not necessary; although defined as such in the text, NO3- and NH4+ are clearer.**

Yes, revised and just used $NO_3^-$ and $NH_4^+$.

**Table 4/5: Define TP and A-P: α values of total phosphorus (TP) and A-P**

Ok, TP and A-P are total phosphorus and available phosphorus, respectively. α values indicate natural logarithm of ratio flower litter addition to non-addition control of different soil indexes (TN, $NO_3^-$-N, $NH_4^+$-N, TP, A-P). See **Line 347-351.**

**Table 6: Mean values (not comparison medium values)**

Sorry for this mistake. We have corrected.

**Comments regarding figures:**

**Figure 1: It is very difficult to identify where the sampling sites are on the map because the elevation shading is so dark (either increase shading transparency or make text and symbols larger)**

Yes, map has been re-drawn with clearer text and symbols. Hope it is qualified.

[Figure]

**"Figure 2: Include the mean (n=X)."**

[Figure]

We have added the mean (n=X) for all the plots and with n value in (b). In (a), the values of sample number are the same (n=20) and we just mentioned in the figure caption. See **Line 684-687**.

**Figure 3: Explicitly state the statistical analysis used (are the bars 95% confidence intervals/SE, or quantiles)? If the whiskers represent SE it seems impossible that the flower litter vs. leaf litter means are significantly different from each other. What are the values (mean, n=X)?**

Ok, we supplied detailed statistical analysis in methods part. In figure 3, box-plots are used to better present the range of data distribution. Bars/whiskers refer to quantiles for comparable settings of all data distribution except extreme outliers (asterisk *). The values (mean, n=X) are also stated by One-way ANOVA. See **Line 279-282**.

**Figures 4 & 5: Define the variables in the figure caption (dissolved inorganic nitrogen (DIN), dissolved organic nitrogen (DON), etc). Mean, n=X. What do the boxes represent? What does deviation from the 0 lines signify (significantly different at what level)?**

Sorry for this unclear display. In fact, they are scatters but not boxes. However, we have to draw them a little bit bigger for the relative smaller error bar, otherwise, both will be overlapped and not well presented. These boxes (scatters) represent α mean values of different indexes in soil N and P pools

after flower litter addition (n=3). It is significantly different at *P*=0.05 level for deviation from the 0 lines. The variables in the figure caption have been defined regarding revised figure and relevant context.

**Figure 6: letters indicate significant differences (at what level, p=0.05)?**

Different lowercase letters indicate significant differences of decomposition rate between litter materials at *P*=0.05 level. It is Fig. 5 in revised version. See **Line 710-713**.

**Comments regarding cited literature**

**"Overall, the authors seem well read on these topics. There are some citations that do not seem relevant to the paper the way it is currently written (for example, findings from tropical, agricultural, and Arctic sites). I think these findings are important because they show flowering litter quality influences soil nutrient status across ecotones, but this needs to be made explicitly clear. The authors also touch on soil organic matter formation/stabilization processes. There is a highly relevant body of literature that could be incorporated to strengthen these points (Kogel-Knaber, Sollins, Cotrfuo, Kleber, etc)."**

Yes, after restructured introduction part, we read carefully about the literatures recommended by referee and also incorporated properly.

 "**I would highly suggest a thorough language editing review is taken before the paper is published**."

Thanks, it has been done regarding your suggestion with attached certificate.

Many thanks to you time!

Jinniu Wang on behalf of all co-authors

To whom it may concern:

This memo is to certify that the paper titled Flower Litters of Alpine Plants Rapidly Affect Soil Nitrogen and Phosphorus in the Eastern Tibetan Plateau has been edited for language by EnPapers, a company dedicated to helping international researchers publish their findings in the best English language journals possible.

Our International paper editing service is performed by a subject expert editor and approved by two senior editors. All our editors are native English speakers.

The certificate is being issued upon the request of the client. If you have any questions, please contact papers@enpapers.com

Signature of the editor representative:

Martin J. Booth

---

## Author Response (AR2)

Dear Editor Dr. Richard,

Many thanks for your kind suggestions! We would like to process the correction of this manuscript entitled "Flower litters of alpine plants affect soil nitrogen and phosphorus rapidly in the eastern Tibetan Plateau" (bg-2016-68), which can be noticed in the revised version with yellow color highlighted for your consideration. Line 90 has been deleted for better reading and understanding. We are looking forward to your reply.

Thanks again for your time, and have a good day.

Jinniu Wang on behalf of all co-authors